# Multiverse Mechanica: A Testbed for Learning Game Mechanics via Counterfactual Worlds

**Robert Osazuwa Ness**[*][†]
Microsoft Research
robertness@microsoft.com

**Ricardo Cannizzaro**[*][‡]
University of Oxford
ricardo@robots.ox.ac.uk

**Yunshu Wu**[*][‡]
UC Riverside
ywu380@ucr.edu

**Lars Kunze**[§]
UWE Bristol
lars.kunze@uwe.ac.uk

## Abstract

We study how generative world models trained on video games can go beyond mere reproduction of gameplay visuals to learning *game mechanics*—the modular rules that causally govern gameplay. We introduce a formalization of the concept of game mechanics that operationalizes mechanic-learning as a *causal counterfactual inference* task and uses the *causal consistency principle* to address the challenge of generating gameplay with world models that do not violate game rules. We present **Multiverse Mechanica**, a playable video game testbed that implements a set of ground truth game mechanics based on our causal formalism. The game natively emits training data, where each training example is paired with a set of causal DAGs that encode causality, consistency, and counterfactual dependence specific to the mechanic that is in play—these provide additional artifacts that could be leveraged in mechanic-learning experiments. We provide a proof-of-concept that demonstrates fine-tuning a pre-trained model that targets mechanic learning. Multiverse Mechanica is a testbed that provides a reproducible, low-cost path for studying and comparing methods that aim to learn game mechanics—not just pixels.

## 1 Introduction

Interactive world models have recently gained attention for their potential to simulate or extend video game experiences (Bruce et al., 2024; Parker-Holder & Fruchter, 2025; Decart et al., 2024; He et al., 2025; Che et al., 2025). SOTA models, typically leveraging deep autoregressive transformer architectures, are trained on large datasets containing sequences of visual frames, user inputs, or internal virtual states produced by a graphics engine. These models of video games (game world models) can generate gameplay sequences that are visually similar to original gameplay; an impressive feat considering modern video games often have cinematic levels of visual complexity.

A key motivation for a focus on video games is the generation of novel gameplay experiences (Gingerson et al., 2024). To this end, the ability to produce high-fidelity visuals is a necessity, as the novel experiences must look the part. But in addition to looking good, the game world model must generate gameplay that is **consistent** with the game's **mechanics** (Gingerson et al., 2024). In simple terms, if the generated gameplay violates game rules or logic, it breaks the gaming experience and, therefore, is not useful, regardless of how good it looks.

Authors of game world models often claim to have learned a game's *mechanics*—rules governing gameplay—through post-hoc observations of generated gameplay from the trained model, which visually demonstrates the mechanics in play. For example, in reference to their *World Models* framework, the authors Ha & Schmidhuber (2018) claim that "by learning only from raw image data collected from random episodes, [their model] learns how to simulate the essential aspects of the game, such as the game logic, enemy behavior, physics ...". Kim et al. (2020) claim their *GameGAN*

---

[*]Equal contribution.
[†]Corresponding author. See Appendix D for author contribution details.
[‡]Work conducted during an internship at Microsoft Research.
[§]Also affiliated with the Department of Engineering Science, University of Oxford, UK.

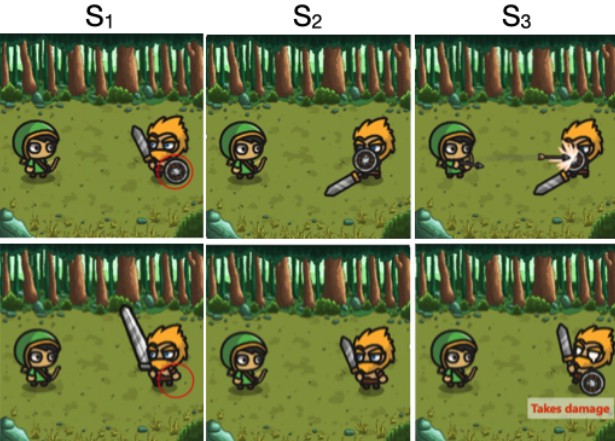

Figure 1: Clips from *consistent contrasts* (3.1.2) sampled at their respective impact frames: each column contains two contrasting gameplay clips where differences are solely attributable to the shield mechanic. Left, middle, and right correspond to parallel world statements $S_1, S_2, S_3$.

model learned the collision and power pellet mechanics of PAC-MAN. Parker-Holder & Fruchter (2025) claimed that consistency was an "emergent" property of Genie 3.

There is a problem with such claims. An *a posteriori* observation that a game-world model has learned a mechanic suggests that, with SOTA architectures, it is *sometimes* possible to learn *some* mechanics—or, more precisely, to learn *representations* that are sufficient to generate gameplay consistent with those mechanics.

However, this does not tell us *a priori* whether it is possible to learn a representation of a mechanic sufficient to reproduce a core element of gameplay. To use a game world model to reliably generate gameplay experiences in practical settings, we need this knowledge before expending on training and deploying a model.

Part of the difficulty is a lack of any formalized notion of what it means to learn a game mechanic. With a formal specification, we could determine if a mechanic is statistically identifiable from training data, and we could evaluate claims that a model has learned a mechanic. Moreover, we could define exactly what it is that we want to identify. The model needs to learn a representation of a mechanic sufficient to reliably reproduce it in generated gameplay, without going as far as reverse-engineering the mechanic's source code; a formal specification would tell us the level of detail required. Lacking this, we cannot know when we have disentangled the mechanic's representation from learning other representations that drive the game's visuals.

In this paper, we address this problem with the following contributions. Firstly, we provide a **causal formalization** of the concept of game mechanics using causal graphical modeling theory. We use this formalization to demonstrate data and inductive biases (in the form of causal graphs) that enable learning a game mechanic. We introduce *Multiverse Mechanica*, a playable game for use as a testbed for evaluating the learning of game mechanics. The game mechanics are implemented with our causal graphical formalization, providing a ground truth for evaluating mechanic learning. It provides causal graphical representations of the mechanic directly to the user for use in evaluation or to supervise mechanic-learning. Multiverse Mechanica is visually simple enough to facilitate inexpensive experimentation, while it enables generalization by using mechanics that are typical of fantasy combat games. Finally, we provide a proof-of-concept for fine-tuning a generative model on a specific game mechanism that leverages these causal representations in its objective function.

## 2 BACKGROUND AND RELATED WORK

**Defining Game Mechanics.** Building on prior definitions (Lo et al., 2021), we define a *game mechanic* as a *modular subset* (Björk & Holopainen, 2004; Schaul, 2013; Thielscher, 2011; Zook & Riedl, 2019) of the *game rules* triggered by specific player/agent interactions (Lundgren & Bjork, 2003; Fullerton et al., 2004), producing transitions in *game state* (Järvinen, 2008; Fabricatore, 2007),

| Level | Example Statement | Description |
|---|---|---|
| 1) Observation | "A warrior with a light weapon *might* equip a shield, block an incoming attack, and take no damage." | A prediction drawn from the distribution of observed gameplay; no variable is set by fiat. |
| 2) Intervention | "*If* a heavy-weapon warrior *were given* a shield, they *would* block an incoming attack and take no damage." | Shield is forced on, overriding its natural dependence on weapon type. |
| 3) Counterfactual | "*Given* a shieldless warrior took damage from an attack, *if* they had instead had a shield and blocked, they *would have* taken no damage—with all else in the scene unchanged." | A specific observed episode is revisited under a contrary-to-fact change, with unaffected variables held *consistent*. |

Table 1: The three causal-hierarchy levels illustrated with the shield mechanic. Each row asks a progressively stronger causal claim about the same gameplay scenario.

including transitions that affect gameplay visuals (Hunicke et al., 2004). These subsets entail causal relations with preconditions and effects, representable as logic, finite-state machines, behavior trees, or transition functions (Zook & Riedl, 2014; Thielscher, 2011; Schaul, 2013; Dormans, 2012; Zook & Riedl, 2019). We formalize this definition in Section 3.2.

**Causal Framing.** We adopt the *causal hierarchy*, which describes three progressively stronger levels of causal reasoning: (level-1) observation, (level-2) intervention, and (level-3) counterfactual (Bareinboim et al., 2022). A level-1 observational statement predicts what *might* happen based on the distribution of observed gameplay, $P(\mathbf{V})$; no variable is set by fiat. A level-2 interventional statement asks what *would* happen if a variable were forced to a value—possibly one that overrides its natural causes—targeting $P(\mathbf{V}_{X=x})$. A level-3 counterfactual statement revisits a specific observed outcome and asks what *would have* happened under a contrary-to-fact change, while holding all unaffected variables are held *consistent* between the observed **world** and the hypothetical one; it targets joint quantities such as $(Y_{X=x}, Y_{X=x'})$ that encode "all-else-equal" comparisons. Here, the term "world" refers to a hypothetical scenario, and we contrast worlds under different hypothetical conditions such as $X = x$ versus $X = x'$. Table 1 illustrates these three levels with concrete examples from the shield mechanic. Further background on the causal hierarchy is provided in Appendix E.1, including an introduction to the above notation in Table 4.

We use causal DAGs and *structural causal models* (SCMs) (Pearl, 2009) to model mechanics as mechanisms (rules) (Bongers et al., 2018). For focusing on a mechanic's variables, we reference *marginalized DAGs* (mDAGs) that preserve causal and interventional semantics while marginalizing others (Evans, 2016). The *causal consistency principle* states that variables not downstream of an intervention retain the same value across *worlds*. (Pearl, 2010; Shpitser & Pearl, 2012); *counterfactual graphs* capture this cross-world consistency compactly (single nodes for shared variables; world-indexed nodes for affected ones) (Shpitser & Pearl, 2012). Construction details appear in Appendix E.5; additional background is in Appendix E.

**Learning game mechanics.** Empirical results highlight the challenges of learning game mechanics. A study performed by Gingerson et al. (2024) highlights the continuing challenge of generating *consistent* gameplay with SOTA architectures—even when generations look plausible, they frequently break the rules of the mechanic. Chen et al. (2025) report similar failures in spatial and numerical consistency, necessitating explicit corrective modules. More broadly, empirical studies of video prediction models show that while they excel in-distribution, they often rely on case-based mimicry and fail under distribution shift, violating simple physical principles (Kang et al., 2024; Riochet et al., 2021).

If we view a mechanic as a latent generative factor, then unsupervised learning of mechanics is provably impossible from video observations of gameplay alone (Locatello et al., 2019) without strong inductive biases (Mitchell, 1980; Wolpert, 1996). Prior work in this area shows that causal representations are at best only partially identifiable from observational data without intervention data or strong causal inductive biases (Spirtes et al., 2000; Bareinboim & Pearl, 2016; Schölkopf et al., 2021). Our work builds on prior work that employs these causal approaches to learning latent generative factors. But to our knowledge, our work is the first to apply this type of causal analysis to

| Descriptions in Causal Logic | Formal Counterfactual Notation | Consistent Contrast Sample Data |
|---|---|---|
| **S₁** All else equal, if the opponent has a light weapon, they may equip a shield; if heavy, they cannot. | $\mathcal{S}_1: P(S_{W=1} = 1, S_{W=0} = 0) \geq \epsilon_1$ | $\mathcal{D}_1 = \{\omega_1, C, (S_{W=1}, B_{W=1}, D_{W=1}, V_{W=1}), (S_{W=0}, B_{W=0}, D_{W=0}, V_{W=0})\}$ |
| **S₂** All else equal, if the opponent has a shield, they may block; if no shield, they cannot. | $\mathcal{S}_2: P(B_{S=1} = 1, B_{S=0} = 0) \geq \epsilon_2$ | $\mathcal{D}_2 = \{\omega_2, C, W, (B_{S=1}, D_{S=1}, V_{S=1}), (B_{S=0}, D_{S=0}, V_{S=0})\}$ |
| **S₃** Given a shield, if a block succeeds then no damage; if it fails, damage occurs. | $\mathcal{S}_3: P(D_{B=1} = 0, D_{B=0} = 1 \mid S = 1) \geq \epsilon_3$ | $\mathcal{D}_3 = \{\omega_3, C, W, S, (D_{B=1}, V_{B=1}), (D_{B=0}, V_{B=0})\}$ |

Table 2: The shield mechanic described as in natural language causal logic (column 1), which are then formalized with counterfactual notation (column 2), where strictly positive probabilities ($\epsilon_i > 0$) indicate bounded uncertainty due to other causal factors in the system. Column 3 shows *consistent-contrast* tuples (each row shares seed $\omega_i$).

the problem of learning game mechanics during training, and generating consistent gameplay from a trained model.

**Datasets, testbeds and environments.** Existing testbeds, datasets, and environments for world models and video prediction largely emphasize intuitive reasoning about real-world Newtonian physics rather than explicitly defined game mechanics. For instance, IntPhys (Riochet et al., 2018) probes intuitive physics by testing whether models respect basic object permanence and motion, while Physion (Bear et al., 2021) provides simulated videos of collisions and stability events to evaluate physical prediction. However, game mechanics can encompass non-realistic "physics," such as spell casting and passing through portals. *Multiverse Mechanica* focuses on a broader set of game mechanics, and contributes a *playable generator* that emits data and artifacts that target learning of a formally defined ground-truth set of mechanics.

## 3 FORMALIZING AND LEARNING A GAME MECHANIC

In this section, we demonstrate the formalization of a game mechanic as well as how we would learn that mechanic from data. Then, in Section 3.2, we provide a general mathematical description of this approach.

### 3.1 ILLUSTRATING EXAMPLE

**The Shield Mechanic** Consider a scene from a stylized 90's fantasy turn-based combat game, where an archer battles a warrior. Like many games in this genre, there is a *shield mechanic*, as shown in Figure 1, where the warrior may raise a shield to block incoming attacks.

#### 3.1.1 FORMALIZING THE SHIELD MECHANIC

How might we describe this shield mechanic in formal causal terms?

**Step 1: Describe the Mechanic with Causal Logic.** We start by completely describing the shield mechanic using a series of causal hypothetical statements of the form "Given preconditions $W$, all else equal, if $X$, then $Y$." Specifically, we focus on level-3 multiverse logic statements that employ conjunctions of conflicting conditions. Table 2 column 1 shows three statements, **S₁**, **S₂**, and **S₃**, that fully describe the shield mechanic.

The columns of Figure 1 correspond to **S₁**, **S₂**, and **S₃**. We could instead use *level-2* interventional statements, which are normally preferred because they are generally testable with experimental data. But the *level-3 parallel world* statements provide an additional constraint in the phrase "all else equal"; that outcomes unaffected by the conditions must remain *consistent* across the clauses. As we will see below, we can use that constraint to operationalize consistency in game generation. Secondly, we can leverage the gaming setting's rare opportunity to generate level-3 data to validate level-3 statements.

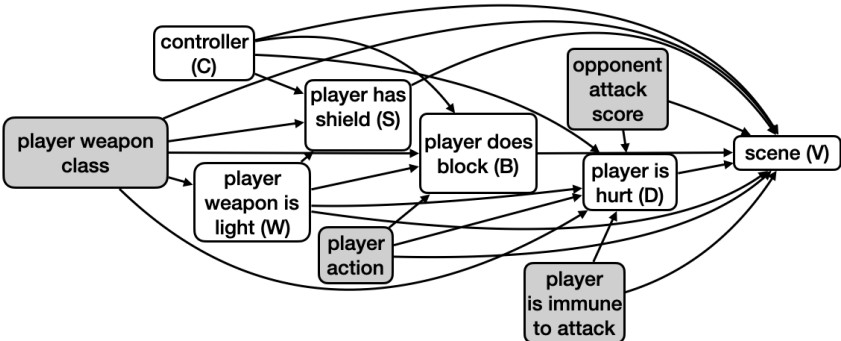

Figure 2: Causal DAG marginalized to focus on the nodes specific to the shield mechanic and their latent confounders (gray nodes)

**Step 2: Rewrite as Counterfactual Expressions.** We can rewrite $\mathbf{S_1}$, $\mathbf{S_2}$, and $\mathbf{S_3}$ as mathematical expressions using counterfactual notation, capturing the contrasts more compactly (see Table 4 for an introduction to probabilistic counterfactual notation). For simplicity, let $W$ denote the weapon type, $S$ indicate whether a shield is equipped, $B$ indicate whether the shield is used to block, and $D$ indicate whether damage occurs. We treat these as binary variables for clarity, without loss of generality. Let $W = 1$ and $W = 0$ denote light and heavy weapon, respectively. $S$, $B$, and $D$, let 1 mean *True* and 0 mean *False*.

We use counterfactual notation to denote variables under the influence of intervention, such that $Y$ under an intervention that sets X to x is written as $Y_{X=x}$. We can formalize the parallel world statements $\mathbf{S_1}$, $\mathbf{S_2}$, and $\mathbf{S_3}$ as shown in column 2 of Table 2, where $\epsilon_i; \forall i \in \{1, 2, 3\}$ denotes strictly positive probabilities ($\epsilon_i > 0$), indicating bounded uncertainty due to other causal factors in the system.

With this, our shield mechanic is described in formal mathematical terms.

**Step 3: Represent Mechanic with Causal Graphs** Let $V$ denote a full clip of gameplay. An outcome, denoted $v$, is a sequence of frames. Let $C$ denote the controller input from a player at the start of the player's turn. Let $G$ denote the full causal DAG for a single turn in the battle (see the full graph in Figure 6 in Appendix F). Let us assume we have access to this DAG, or that we could create it using knowledge of the game structure, analyzing causal dependence in the game's code (Winskel, 1986; Aho et al., 2006), or by applying causal discovery methods (Glymour et al., 2019). The variables implicated in our description of the shield mechanic are $Z = \{C, W, S, B, D, V\}$. The causal DAG $G$ is quite large, so we derive the mDAG $G^{\mathcal{M}}$ that zooms in on $Z$ (Figure 2) by marginalizing out the variables not in $Z$ (see Appendix E.3 for a description of the algorithm). Next, we can combine the mDAG with each counterfactual expression in the mechanic's description to construct the counterfactual graphs in Figure 3.

The counterfactual graphs in Figure 3 encode a representation of *causal consistency*—variables that are not downstream of interventions and thus are consistent across worlds are unique, while inconsistent variables have nodes indexed by each world. Thus, the counterfactual graphs are representations of the shield mechanic that explicitly describe what should remain consistent when generating gameplay depicting the shield mechanic.

### 3.1.2 GENERATING SHIELD MECHANIC DATA

We can generate level-3 parallel world data consistent with $\mathcal{S}_1$, $\mathcal{S}_2$, and $\mathcal{S}_3$ by creating parallel runs with identical initial conditions and random seeds. We can intervene separately in each run, producing clips of parallel *virtual* worlds that differ only in their respective interventions. We call the tuple of these clips, combined with the outcomes of other mechanic-related variables under these interventions and their shared initial condition/seed, *consistent contrasts* . Figure 1 illustrates gameplay clips from the contrasts.

Let $\omega_1$, $\omega_2$, and $\omega_3$ represent distinct sets of random seeds and initial conditions. Let $C$ represent the controller input from the player. Let $V_{X=x}$ represent a video clip of gameplay under an intervention

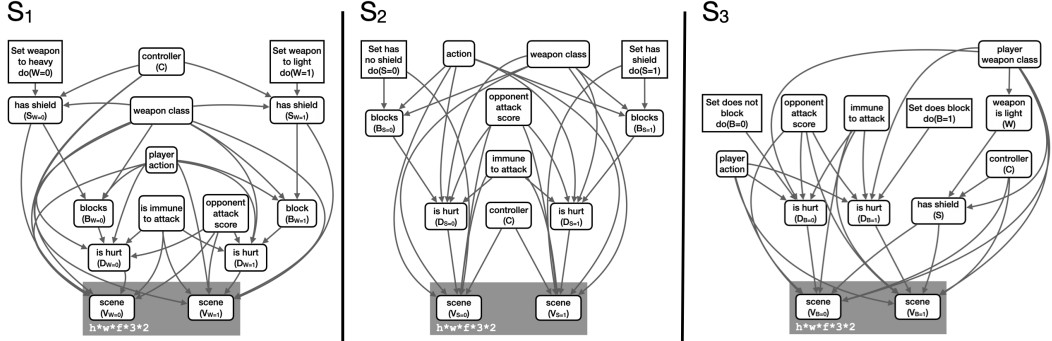

Figure 3: Counterfactual graphs for the shield mechanic. The video variables $V_{X=x}$ have shape height(h)*width(w)*frames(f)*channels(3)*worlds(2).

that sets $X$ to $x$. Reading consistency from the graphs in Figure 3, we see $W_{S=1} = W_{S=0} = W$, $W_{B=1} = W_{B=0} = W$, and $S_{B=1} = S_{B=0} = S$. Let $\mathcal{D}_1$, $\mathcal{D}_2$, and $\mathcal{D}_3$ represent samples of consistent contrasts for $\mathbf{S_1}$, $\mathbf{S_2}$, and $\mathbf{S_3}$ respectively, as shown in Table 2 column 3.

### 3.1.3 LEARNING THE SHIELD MECHANIC FROM DATA

Let $P_1 = P(S_{W=1}, S_{W=0})$, $P_2 = P(B_{S=1}, B_{S=0})$, and $P_3 = P(D_{B=1}, D_{B=0}, S)$ denote the distributions constrained by $\mathcal{S}_1$, $\mathcal{S}_2$, and $\mathcal{S}_3$. We can estimate these distributions through repeated sampling of consistent contrasts $\mathcal{D}_1$, $\mathcal{D}_2$, and $\mathcal{D}_3$, then averaging over the sampling distributions to obtain the sampling distributions $\hat{P}_1$, $\hat{P}_1$, and $\hat{P}_3$ (see Appendix E.6). In each case $\hat{P}_i$ converges almost surely to $P_i$. This provides a precise operationalization of what it means for a generative model to learn the shield mechanic: learning constraints $\{\mathcal{S}_1, \mathcal{S}_2, \mathcal{S}_3\}$ on distributions $\{P_1, P_2, P_3\}$ or modeling $\{P_1, P_2, P_3\}$ directly.

However, in the canonical case of training game world models, we assume that only the controller inputs and the video outputs are observed during training. Here, the inference of $\hat{P}_i$ becomes a task of unsupervised learning of a latent vector $\mathcal{D} \setminus \{C, V_{X=x}, V_{X=x'}\}$ using $\{C, V_{X=x}, V_{X=x'}\}$ as features, where $V_{X=x}, V_{X=x'}$ is a vector of shape 2 * frame height * frame width * 3 RGB channels * number of frames. Without further assumptions, disentangling the components of $\mathcal{D} \setminus \{C, V_{X=x}, V_{X=x'}\}$ is generally infeasible. However, the counterfactual graphs in Figure 3 already disentangle these variables for us. Using these, the problem reduces to training a latent variable model.

## 3.2 FORMAL FRAMEWORK FOR GAME MECHANICS

We assume a causal DAG $G = (\mathsf{V}, \mathsf{E})$ for a single step of gameplay with $\mathsf{V} = \{C_t, X_t, C_{t+1}, X_{t+1}, V_t, V_{t+1}\}$, and typical edges $X_t \to C_t$, $X_t \to X_{t+1}$, $C_t \to X_{t+1}$, $X_t \to V_t$, $X_{t+1} \to V_{t+1}$. For a given mechanic, we restrict to the variable subset $M = \{C_t, X_t^M, X_{t+1}^M, V_{t+1}\}$, $X_t^M \subseteq X_t$, $X_{t+1}^M \subseteq X_{t+1}$, and form the marginalized DAG $G^M$ by marginalizing variables outside $M$ while preserving interventional semantics (Appendix E.3).

We work on a probability space with sample space $\Omega$. Each variable $Z \in M$ is a measurable mapping $Z : \Omega \to \mathcal{X}_Z$, and each counterfactual $Z_{X=x} : \Omega \to \mathcal{X}_Z$, where $x \in \mathcal{X}_X$ is an intervention value in the state space of $X$. The ground-truth SCM, consistent with $G^M$, induces the family

$$\mathcal{F}(M) = \Big\{ P\big(V_{t+1, X=x}, Z_{X=x} \,\big|\, E\big) \,:\, X, Z \in M, \ x \in \mathcal{X}_X, \ E \in \sigma(M) \Big\},$$

where $\sigma(M)$ denotes the $\sigma$-algebra generated by the variables in $M$ (in practice, $E$ can be any measurable predicate on $M$, e.g., $S{=}1$ for "has shield").

We formalize a mechanic as the tuple $\langle G^M, \mathcal{M} \rangle$, where $\mathcal{M} = \{S_1, \ldots, S_k\}$ and each constraint $\mathcal{S}_i$ : $P\Big(\bigwedge_{j=1}^{m_i} Y_{X=x_j} = y_j \,\Big|\, E\Big) \geq \epsilon_i$ binds counterfactuals of variables in $M$ under interventions on $X \in M$, with $x_j \in \mathcal{X}_X$, $E \in \sigma(M)$, and $\epsilon_i \in (0,1]$ (allowing non-deterministic relations due to factors outside $M$). For each $\mathcal{S}_i$ we denote the targeted distribution by $P_i$ (e.g., $P_1 = P(S_{W=1}, S_{W=0})$ in the shield example).

**Data and Estimation.** A consistent-contrast dataset for $\mathcal{S}_i$ of size $N$ is

$$\mathcal{D}_i^{(N)} = \left\{ \left(V_{X=x_1}(\omega_n), \ldots, V_{X=x_{m_i}}(\omega_n)\right) \ : \ \omega_n \in \Omega, \ n = 1, \ldots, N \right\},$$

optionally restricted to $\omega_n$ satisfying $E$. Let $\hat{P}_i^{(N)}$ be the empirical distribution induced by $\mathcal{D}_i^{(N)}$. Under i.i.d. sampling of seeds $\omega_n$, $\hat{P}_i^{(N)} \xrightarrow{\text{a.s.}} P_i$. For each $\mathcal{S}_i$ (and associated $P_i$), we construct a *counterfactual graph* $G_i^{M,\text{cf}}$. In partially observed settings (video + controller only), $G_i^{\text{cf}}$ specifies which latent variables are shared across worlds and which differ, reducing estimation to a well-posed latent variable problem aligned with the counterfactual graph's structure.

# 4 MULTIVERSE MECHANICA: A PLAYABLE TESTBED FOR LEARNING MECHANICS

We introduce *Multiverse Mechanica*, a fantasy-style battle game designed as a testbed for learning game mechanics. Unlike static datasets, Multiverse Mechanica is a playable game that emits the artifacts required to study and evaluate whether models capture the game's mechanics—not just gameplay visuals. Its design integrates three innovations: (i) native support for level-3 parallel-world contrasts with consistency under the same $\omega$; (ii) per-mechanic mDAGs $G^{\mathcal{M}}$, parallel world and counterfactual graphs, and specifications of $\mathcal{M}$; and (iii) explicit visual grounding, where stance and scene variables are rendered into pixels. The full codebase for Multiverse Mechanica is publicly available to support reproducible research on mechanic learning.[1]

## 4.1 GAME OVERVIEW

Each episode consists of a pre-battle setup (character and equipment selection, random assignment of elemental buffs (e.g., fire, ice) followed by turn-based combat. The player occupies the left side of the screen, and the enemy occupies the right. On the player's turn to attack, a timing-based interaction yields an attack score; the enemy's turn samples an analogous attack score. Outcomes depend on weapons, defenses, the attack score, and buffs. See Appendix G for additional details.

## 4.2 IMPLEMENTED MECHANICS (V1.0)

Version 1.0 of Multiverse Mechanica includes the following mechanics, each with associated $G^{\mathcal{M}}$ and parallel-world data sufficient to estimate $\mathcal{M}$. The **shield mechanic** focuses on equipping and blocking with a shield, as discussed in Section 3.1. In the **elemental immunity mechanic**, "elemental" attributes (e.g., fire and ice) govern immunity and vulnerability to attacks. The **weapon range mechanic** governs melee vs. ranged combat. The **spell-casting mechanic** governs five submechanics that allow players to give themselves an advantage in battle (e.g., gain increased attack power, dodge ability), or their opponent a disadvantage (e.g., disarm them or lower their defense)—projectiles, self-levitation, enemy-levitation, self-transformation, and enemy-transformation. See Appendix F for detailed descriptions, including causal formalizations, DAGs, and illustrations.

## 4.3 DESIGN DECISIONS

**Mechanic-Specific Game Systems.** Each mechanic is implemented with a unique instance of a *game system* (Nystrom, 2014; Gregory, 2018), independent of the others. This ensures parity between the mechanic and the code logic. Full game system details are given in Appendix G.1.1.

**Impact Frames and Visual Conventions.** We designed the game such that each turn contains an *impact frame*—the most visually and mechanically expressive phase of an interaction (e.g., the precise moment when an attack lands). Impact frames are not defined by fixed time-points; instead, they are triggered by specific game state configurations. See Appendix G.2 for details.

**Simple yet Information Dense Visuals.** To facilitate rapid, inexpensive experimentation, we focus on the ability to run experiments with episodes that have a minimal number of frames. To this

---

[1] https://github.com/ricardocannizzaro/multiverse-mechanica

end, we use a simple art style that aligns with the representational biases of pretrained vision models (luciI, 2024; Zhang et al., 2023). A complete attack–reaction animation plays after player input at turn start, permitting cheaper architectures that model multi-frame action sequences instead of one frame per action. Animation conventions emphasize dynamic information, such as speed lines ("zip ribbons") to depict fast motion, trajectory lines for projectiles, curved swipes for melee attacks, and burst lines and explosion visual effects for collisions or blocked strikes (McCloud, 2020; Eisner, 2008; Cohn, 2013). In Section 5, we highlight this ability by limiting our analysis to single time-point snapshots at the impact frame, chosen as the impact frame of the clip. The images in Figure 1 are all sampled at their respective impact frames.

## 4.4 DATA GENERATION

Multiverse Mechanica is not a dataset but a generator. To generate data, an automated agent repeatedly plays the game to produce clips. Users can select a number $N$ of generations. The generation process can randomly generate $N$ clip examples, constituting level-1 data. The user can also specify interventions on specific game state variables and generate $N$ clip examples where those interventions are applied, constituting level-2 data. Finally, the user can specify interventions and assign them to multiple game instances with a shared "$\omega$" (same random seed and initial conditions) and generate $N$ consistent contrasts (tuples of clips), constituting level-3 data. Each mechanic has presets for level-2 and level-3 generation. Each clip is a 512x512 MP4 video averaging 4 seconds at 50 FPS. Each generated example is a tuple consisting of a clip, controller inputs, game-state variable outcomes, and a random seed for reproducibility. See Appendix H for additional details related to data generation.

**Summary.** Multiverse Mechanica provides a compact yet expressive testbed for studying whether generative models can capture mechanics. Its design couples causal structure with visual grounding, leverages art and animation conventions for low-cost training, and enables reproducible creation of parallel-world contrasts.

## 5 PROOF-OF-CONCEPT: LEARNING A MECHANIC WITH MULTIVERSE FINE-TUNING

In this section, we provide a proof-of-concept that illustrates how the proposed causal graphical approach to identifying and learning game mechanics can motivate a new world model training strategy. Given realistic player action policies, real-world game engines could be reconfigured into simulators that generate consistent contrasts depicting the mechanics they implement. We could then treat the visual information of what remains consistent across the pair as supervision in a contrastive learning objective. Using this objective, we could fine-tune a world model to improve its causal consistency; a method we call *multiverse alignment*. The initial training phase learns a rough representation of the mechanic, then multiverse fine-tuning refines that representation into one capable of producing causally consistent depictions of the mechanic. We demonstrate this idea using consistent contrasts generated with Multiverse Mechanica, which generates consistent contrasts by design.

There is no established metric for causal consistency of full video roll-outs. Practice is to compare event-aligned frames—same event label, possibly different timestamps—with image similarity (PSNR, SSIM) (Wang et al., 2004; Wolf et al., 2009). This is orthogonal to within-video temporal generation. We thus evaluate our proof-of-concept by applying these metrics to multiverse fine-tuning of a pre-trained conditional image diffusion model targeting event-aligned frames.

Concretely, we generated contrastive pairs spanning the full set of mechanics (i.e., shields, elemental immunity, weapon range, and spell-related submechanics), as discussed in Section 4.2. Each pair consists of two video roll-outs that depict a complete interaction: an attacker executes an attack animation, and the reactor produces the corresponding response. For every roll-out, the simulator also records a dictionary of outcome values for the game-state variables that determine the events shown in the video. For each contrastive pair, we extract the impact frame (Section 4.3) from each roll-out, yielding a paired set of event-aligned images. We then fine-tune the conditional image diffusion model OpenJourney-v4 (PromptHero, 2022) on these impact-frame pairs. To condition

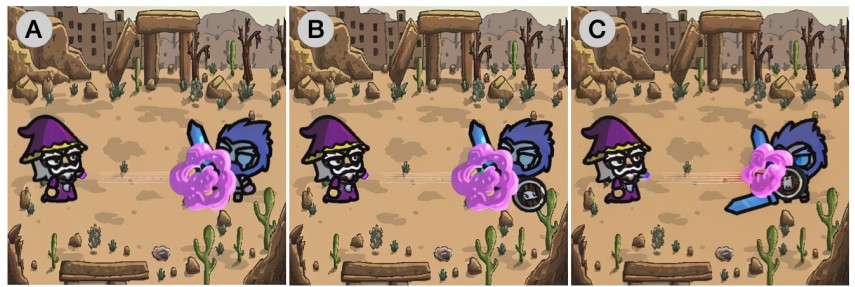

Figure 4: Counterfactual samples from the multiverse-aligned diffusion model (shield mechanic). Non-mechanic content is preserved across interventions. (A) Held-out factual image: wizard attacks warrior. (B) $V_{S=1,B=0}$: shield present, no block. (C) $V_{S=1,B=1}$: shield and block active (note: minor sword aliasing artifact).

the model, we convert the associated game-state dictionary into a textual description that is provided as input.

Conceptually, we consider an image $V_{X=x}$ as a snapshot of a simulated roll-out from the game with mechanic-related state settings $X = x$ (as affected by controller input). The simulation has a random seed $\omega$, a random draw whose value proxies for the effect that variables, which are not in or affected by $X$, have on $V_{X=x}$. A consistent contrast is a paired set of images $(V_{X=x_0}, V_{X=x_1})$ that share a random seed $\omega$. The shared seed enforces causal consistency; the only differences between $V_{X=x_0}$ and $V_{X=x_1}$) are due to different interventions on $X$.

We can view the conditional image diffusion model as a model of how the simulation culminates in snapshot $V_{X=x}$. With timesteps $t \in [0, T]$, $Z_T$ is the noisy latent that initializes the denoising process, and $X$ is the conditioning variable used to guide that process. Starting from completely random $Z_T$, one denoising step conditioned on $X$ yields $Z_{T-1,X=x}$. Iterative denoising from (counting down T to 0) culminates in $Z_{0,X=x}$, which is then decoded into $V_{X=x}$.

**Multiverse alignment** is motivated by viewing $Z_T$ as an analog of $\omega$: both determine what varies in the image given $X$. To reflect the shared $\omega$ in a consistent contrast, we constrain the $Z_T$ in each trajectory to agree across the pair; the objective modifies standard diffusion training to enforce this.

Conditioned on controller input $c$, the *denoiser* $\epsilon_\theta(\cdot)$ iteratively transforms the shared seed $\omega$ into a clean latent $Z_{0,X=x_j}$ that decodes into the frame $V_{X=x_j}$. The **multiverse alignment** loss has two components:
$$\mathcal{L} = \lambda_1 \mathcal{L}_1 + \lambda_2 \mathcal{L}_2, \quad \lambda_1, \lambda_2 \geq 0, \; \lambda_1 + \lambda_2 = 1.$$

$\mathcal{L}_1$**: seed-consistency loss.** $\text{Abduct}_\theta$ traces $Z_{0,X=x_j}$ and $c_j$ backward via $\epsilon_\theta(\cdot)$ through the noise schedule to estimate $\hat{\omega}_j$ (deterministic inversion; see Appendix J.10.5). The loss enforces agreement between abducted seeds:
$$\mathcal{L}_1(Z_{0,X=x_0}, c_0, Z_{0,X=x_1}, c_1) = \left\| \text{Abduct}_\theta(Z_{0,X=x_0}, c_0) - \text{Abduct}_\theta(Z_{0,X=x_1}, c_1) \right\|_2^2.$$

*Remark.* We use deterministic diffusion inversion to estimate $\omega$ and penalize differences between abducted seeds.

$\mathcal{L}_2$**: structure-alignment loss.** Let $S \subset \{1, \ldots, T\}$ be a subset of high-noise timesteps. For each $t \in S$, the denoiser predicts noise $\epsilon_\theta(Z_{t,X=x_j}, t, c_j)$. We align predictions across the contrast:
$$\mathcal{L}_2 = \sum_{t \in S} \left\| \epsilon_\theta(Z_{t,X=x_0}, t, c_0) - \epsilon_\theta(Z_{t,X=x_1}, t, c_1) \right\|_2^2.$$

*Remark:* Aligning high-noise steps enforces global identity; mechanic-specific differences emerge in later steps.

**Evaluation Results.** Figure 4 shows qualitative counterfactual generations: the model preserves non-mechanic-related content while toggling the targeted shield mechanic. From an input "factual" image $v$ with $X = x$ we abduct the exogenous seed $\omega$, then generate a counterfactual image $v'$ with $X = x'$ while keeping $\omega$ fixed, sampling from $P(V_{X=x'} \mid X = x, V = v)$. This provides visual evidence that the fine-tuned model has learned a representation that can generate causally consistent aspects of the mechanic, not merely pixels.

To quantify the contribution of each loss component, we conduct an ablation study over four configurations—standard diffusion fine-tuning (baseline), $\mathcal{L}_2$ only, $\mathcal{L}_1$ only, and the full loss $\mathcal{L}_1+\mathcal{L}_2$—evaluated against engine-rendered ground-truth counterfactuals (Table 3). Reconstruction metrics compare inverted images to their inputs; counterfactual (CF) metrics compare generated counterfactuals to ground truth; and *Exo MSE* measures $\|\hat{\omega}_0 - \hat{\omega}_1\|^2$, the squared distance between abducted seeds of a consistent-contrast pair.

The results divide by the presence of $\mathcal{L}_1$. Without it, Exo MSE remains high ($\approx 0.79$); adding $\mathcal{L}_2$ alone is indistinguishable from the baseline. Including $\mathcal{L}_1$ reduces Exo MSE by nearly an order of magnitude ($\approx 0.09$), enforcing tight seed alignment at a cost of some pixel-level fidelity. The structure-alignment term $\mathcal{L}_2$ complements $\mathcal{L}_1$: adding it improves reconstruction SSIM from 0.275 to 0.429 and counterfactual SSIM from 0.129 to 0.196 without degrading seed alignment. CLIP-based evaluation corroborates this: CF Transfer CLIP is substantially higher with the full loss ($\approx 24$ vs. $\approx 17$ for $\mathcal{L}_1$ alone) (Hessel et al., 2021; Radford et al., 2021), and image-level CLIP scores (image–text: 24.42, image–image: 0.898) confirm that the procedure preserves input-side fidelity.

Table 3: Ablation over loss components evaluated against engine-rendered ground-truth counterfactuals. A horizontal rule separates configurations without $\mathcal{L}_1$ (top) from those with it (bottom).

| | Reconstruction | | Counterfactual | | |
|---|---|---|---|---|---|
| **Configuration** | PSNR↑ | SSIM↑ | PSNR↑ | SSIM↑ | Exo MSE↓ |
| Diffusion only | 19.54 | 0.776 | 13.58 | 0.358 | 0.793 |
| w/o $\mathcal{L}_1$ | 19.42 | 0.769 | 13.55 | 0.355 | 0.795 |
| w/o $\mathcal{L}_2$ | 13.35 | 0.275 | 11.01 | 0.129 | 0.087 |
| Full ($\mathcal{L}_1+\mathcal{L}_2$) | 14.02 | 0.429 | 11.04 | 0.196 | 0.092 |

## 6 SCALABILITY & LIMITATIONS

Multiverse Mechanica provides a simplified setting for experiments: a 2D turn-based domain. The logic of the causal formalization generalizes in principle to continuously interactive 3D environments, continuous-valued interventions, and temporal dynamics. However, further work is needed to analyze how to formalize more complex mechanics that arise in such settings. Furthermore, the development of new metrics for evaluating causal consistency across entire roll-outs remains an open challenge.

The proof-of-concept presented here is limited. For simplicity, focused on modeling single impact frames rather than full video roll-outs; we did this with an image model rather than a standard world model architecture. The seed-consistency loss $\mathcal{L}_1$ requires tracking gradients through every step of the DDIM inversion trajectory, making it substantially more expensive than standard diffusion training.

The same causal-consistency formulation extends to other controllable simulators and VFX pipelines; we discuss pathways for transferring these ideas to more realistic settings in Appendix L. *Multiverse Mechanica* serves as a best-case benchmark with explicit contrasts, whose insights can inform models trained under weaker supervision.

## 7 CONCLUSION

We formalized game mechanics as causal counterfactual inference and introduced *Multiverse Mechanica*, a playable testbed that emits parallel-world contrasts and per-mechanic causal graphs. Building on this foundation, we proposed a multiverse-alignment objective and demonstrated a proof-of-concept fine-tuning that learns targeted mechanics. Together, these components provide a reproducible path to assessing whether world models learn *mechanics—not just pixels*.

ACKNOWLEDGMENTS

This research was supported in part by the Australian Defence Science and Technology Group. Lars Kunze's RAi UK Enterprise Fellowship is supported by EPSRC [grant number EP/Y009800/1].

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

## A    USE OF LARGE LANGUAGE MODELS (LLMS)

GPT-5 was used for editing and polishing during the writing of this paper. The LLM was not used to make any technical or scientific contributions to the paper writing process; for example: writing the literature review or background sections, creating citations, or analyzing data.

## B    ETHICS STATEMENT

There are no ethics concerns raised by this paper.

## C    REPRODUCIBILITY STATEMENT

To support full reproducibility of this work, we make the following resources publicly available:

- The complete implementation of Multiverse Mechanica, a research testbed for evaluating learning of game mechanics, including the game engine, causal-graph construction utilities, parallel-world data generation pipeline, and proof-of-concept training scripts, is publicly available at `https://github.com/ricardocannizzaro/multiverse-mechanica`.
- The proof-of-concept dataset used in Section 5, including mechanism-specific graphical artifacts and level-3 parallel-world contrasts, is accessible via links provided in the Multiverse Mechanica repository.

### C.1    PROOF-OF-CONCEPT DATASET

Here, we describe the dataset used for the proof-of-concept training presented in Section 5, used to learn the **shield gameplay mechanic**.

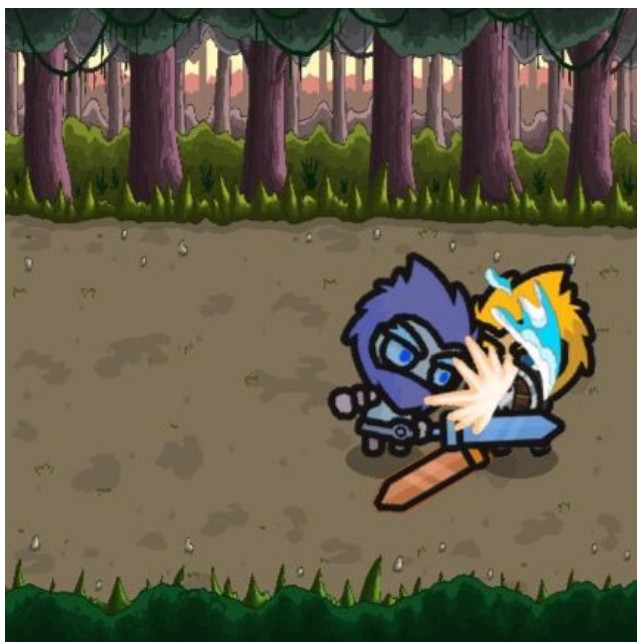

Figure 5: Impact frame generated by Multiverse Mechanica for a world in a parallel world tuple.

### C.1.1 GENERATION

We generate a level-3 dataset, as described in Section 4.4. We automatically derive parallel world interventions based on the full game DAG, targeting the variables relevant to the parallel world contrast statements for the shield mechanic. The set of derived interventions define multiple parallel world tuples, enumerated such that we sufficiently cover the support of joint distribution of the mDAG induced by the query variables associated with the level-3 parallel world statements for the mechanic.

We assign these interventions to multiple game instances with a shared "$\omega$" (same random seed and initial conditions) and generate $N = 1000$ consistent contrasts, constituting level-3 data. Since we are training on text and images (i.e., not gameplay clips), for each we generate only a subset of training artifacts consisting of these two modalities: 1) the impact frame as a 512x512 PNG image, and 2) the game-state variable outcomes (converted into a caption in the pre-processing method, outlined below).

### C.1.2 PRE-PROCESSING

To align our artifact modalities with the text-to-image latent diffusion architecture used in our proof-of-concept, we must perform a pre-processing step to convert from the game-state variable outcomes—a dictionary of variable-name/value pairs—to a text caption. We implement a templated captioning process for each parallel world tuple, by which plain-text captions are deterministically constructed based on the values of the game-state variables.

Consider, for example, the following impact frame image and game-state variable artifacts generated by Multiverse Mechanica for one of the worlds in a parallel world tuple.

For this world, Multiverse Mechanica has generated the impact frame image shown in Figure 5 and the following game-play state:

```
gameplay_state = {
    'background': 'forest',
    'player action': 'melee attack',
    'player class': 'warrior',
    'player does block': False,
    'player element': 'ice',
```

```
    'player has shield': False,
    'player is hurt': False,
    'player is immune to attack': False,
    'player weapon': 'short sword',
    'player weapon class': 'light',
    'player weapon element': 'ice',
    'player weapon is light': True,
    'player weapon range': 'melee',
    'opponent action': 'defend',
    'opponent class': 'warrior',
    'opponent does block': True,
    'opponent element': 'none',
    'opponent has shield': True,
    'opponent is hurt': False,
    'opponent is immune to attack': False,
    'opponent weapon': 'short sword',
    'opponent weapon class': 'light',
    'opponent weapon element': 'fire',
    'opponent weapon is light': True,
    'opponent weapon range': 'melee',
}
```

Given this game-play state, our pre-processing step constructs the following caption:

> "A 1-on-1 battle between two warriors. The warrior on the left has a ice element buff. The warrior on the left's weapon is a short sword with a ice buff. The warrior on the left is launching a melee attack. The warrior on the right's weapon is a short sword with a fire buff and they have a shield. The warrior on the right is blocking with their shield."

The captions generated by the pre-processing step is combined with the generated image to thus create training data suitable for text-to-image latent diffusion architecture used in our proof-of-concept.

## D    AUTHOR CONTRIBUTIONS

We follow the CRediT taxonomy to describe author contributions. Robert Osazuwa Ness led the project, including conceptualization, methodology, and formal analysis, and wrote the original manuscript draft; he also supervised the work, coordinated the project, and led revisions. Ricardo Cannizzaro developed the Multiverse Mechanica codebase and testbed over an extended period, including environment design, combining causal abstractions with game logic, evaluation/benchmarking tooling, dataset generation pipelines, and supporting analysis scripts; he also contributed to manuscript revisions and the reviewer response. Yunshu Wu developed and implemented the proof-of-concept multiverse fine-tuning approach (*Multiverse Alignment*), including the L1 seed consistency loss and L2 structural-alignment loss formulations, experimental codebase, metric selection and implementation, and experiments/ablations; she also contributed text and edits to the manuscript, reviewer feedback, and appendix. Lars Kunze provided supervisory support and manuscript feedback.

## E    BRIEF PRIMER ON CORE CAUSAL CONCEPTS

### E.1    THE CAUSAL HIERARCHY

The *causal hierarchy* describes three levels of statements that employ causal logic: (level-1) observation, (level-2) intervention, and (level-3) counterfactual (Bareinboim et al., 2022). A predictive statement, such as "The scene would generate in *this* way," is a level-1 observational statement. A level-2 interventional statement targets outcomes under hypothetical interventions, such as "If an enemy were placed here (e.g., in a place it would not naturally appear), then the scene would generate in *this* way." A level-3 counterfactual statement considers observed outcomes and how those outcomes might have been different under hypothetical interventions. For example: "Given there

| Variable / Operator | Description |
|---|---|
| $P(X)$ | The probability distribution for random variable X |
| $P(X \mid Z{=}z)$ | As above, but conditioned on observation $Z{=}z$ |
| $\mathrm{do}(X{=}x)$ | An intervention taken to fix variable X to value x. Interrupts sampling from its conditional probability distribution given parents in the causal DAG: $P(X\|Pa(X))$. |
| $P(Y_{X=x} \mid Z{=}z)$ | Probability distribution for Y where an intervention is taken to fix variable X to value x $\big($i.e., $\mathrm{do}(X{=}x)\big)$ (optionally conditioned on observation $Z{=}z$) |
| $\mathbf{S_i}, \mathcal{S}_i$ | The i$^{\text{th}}$ contrast statement, in causal terms and counterfactual notation, respectively |
| $\mathcal{D}_i$ | The i$^{\text{th}}$ *consistent-contrast* tuple, defining two worlds sharing a common seed $\omega_i$ |
| $X_t, C_t, V_t$ | Game data at frame $t$: game state $X_t$, player controller input $C_t$, and video frame (image) vector $V_t$. |

Table 4: Key probability distributions, variables and operators used in the adopted formalism.

was no enemy and the scene generated that way, if an enemy were placed here, the scene would generate in *this* way."

Formally, level-$k$ statements describe events in the sample space of a level-$k$ distribution (Bareinboim et al., 2022). Level-$k$ data can be viewed as samples from such a distribution, and with i.i.d. sampling, the empirical distribution converges to the true sampling distribution. Ordinary gameplay logs are level-1 data; data where variables are artificially fixed before sampling (as in experiments) is level-2 data. In most settings, level-3 data does not exist due to the *fundamental problem of causal inference*—it is impossible to observe outcomes for the same effect variables across worlds (Holland, 1986). In this work, we leverage the video game setting's ability to observe level-3 data across *virtual worlds*, i.e., multiple instances of a game run with shared initial conditions.

Models also align with this hierarchy. A causal DAG is a level-2 model: when combined with a generative model, it encodes the family of interventional distributions over the DAG's variables. A structural causal model (SCM) is a level-3 model: it additionally encodes the family of counterfactual distributions over the DAG's variables (Pearl, 2009). Level-$k$ data is sufficient to identify level-$k$ models, but the *causal hierarchy theorem* states that level-$k$ statements cannot, in general, be inferred from data below level $k$ (Bareinboim et al., 2022).

Thus, we follow the three-level hierarchy (Bareinboim et al., 2022): (L1) *observational* questions about $P(\mathbf{V})$; (L2) *interventional* questions about $P(\mathbf{V} \mid \mathrm{do}(X{=}x))$; and (L3) *counterfactual* (parallel-world) questions about joint outcomes under conflicting interventions, e.g., $(Y_{X=x}, Y_{X=x'})$. Level-3 quantities encode "all-else-equal" comparisons across worlds.

### E.2    SCMs AND RULES

An SCM $\mathcal{M} = (\mathbf{U}, \mathbf{V}, \mathbf{F}, P(\mathbf{U}))$ specifies exogenous factors $\mathbf{U}$, endogenous variables $\mathbf{V}$, structural assignments $\mathbf{F}$, and a distribution over $\mathbf{U}$ (Pearl, 2009). In our context, $\mathbf{F}$ plays the role of executable rules; see Bongers et al. (2018) for SCMs as formal rule systems.

### E.3    MARGINALIZATION OF CAUSAL DAGs

We implement Evans (2016) approach to DAG marginalization. Given a causal DAG and a set of nodes to marginalize out of the DAG, Evans (2016) creates a marginalized DAG with *hyper-edges* that represent the footprint of latent common causes. Directed edges encode parent–child relationships among observed variables as usual. Hyper-edges encompassing a set of observed nodes, indicate that all nodes in the set share an unobserved exogenous influence. For example, a hyper-edge touching nodes $X, Y, Z$ represents the fact that there is some unobserved variable that acts as a common cause to $X, Y, Z$ simultaneously. The mDAG maintains the causal interpretation, intervention model, and implications to conditional independence as the full DAG.

While mDAGs are convenient for reasoning about marginalized structures, we require a fully explicit DAG representation for generating parallel-world and counterfactual graphs, as well as for using standard graph serialization. We therefore modify the algorithm such that, for each hyperedge, a

d-separating sets of shared ancestors that entail the hyperedge are explicitly added back into the model. This implements the expansion of the canonical mDAG by making the latent causes explicit expansion as discussed in Evans (2016).

### E.4    MULTIVERSE REASONING.

With *multiverse* counterfactual reasoning, we envision one world where observed outcomes occurred, and separate "parallel" worlds where hypothetical interventions lead to outcomes that differ from the observed outcomes Shpitser & Pearl (2012). For example, consider the statement "Given there was no enemy and the scene generated *that* way, if an enemy were placed here, the scene would generate in *this* way". With this statement, we can envision two parallel worlds, one where there was no enemy and the scene generated *that* way, and one where there was an enemy and the scene generated *this* way. Level-3 data, therefore, is a data tuple representing outcomes across parallel worlds. The "fundamental problem of causal inference" (Holland, 1986) is that level-3 data is unobservable—in real world settings, it is impossible to observe potential outcomes for the same variable across parallel worlds. A key insight of our work is that in *virtual world* settings, the data can exist by creating parallel instances of the same world with the same initial conditions.

### E.5    COUNTERFACTUAL GRAPHS AND CONSISTENCY

Counterfactual graphs are based on *parallel world graphs*. The parallel world graph clones a causal DAG across parallel worlds and uses graph surgery to represent hypothetical conditions in certain worlds (Shpitser & Pearl, 2012). Variables that are *not* descendants of the intervention share a single node across worlds (consistency); variables affected by the intervention are duplicated and indexed by world. This graph encodes which quantities must remain identical across worlds and which may differ, providing a compact template for supervision and evaluation. The counterfactual graph collapses nodes that must be consistent across worlds into single nodes, creating a graph that (unlike the parallel world graph) encodes conditional independence across parallel worlds. In this work, we formalize the concept of game mechanics such that, for a given mechanic, we can generate a set of parallel world graphs and counterfactual graphs that explicitly encode its structure and which variables should remain consistent while the mechanic is in play. These can be used in training alongside the level-3 gameplay data.

### E.6    ESTIMATING COUNTERFACTUAL DISTRIBUTIONS

In Section 3.1.3, we let $P_1 = P(S_{W=1}, S_{W=0})$, $P_2 = P(B_{S=1}, B_{S=0})$, and $P_3 = P(D_{B=1}, D_{B=0}, S)$ denote the distributions constrained by $\mathcal{S}_1$, $\mathcal{S}_2$, and $\mathcal{S}_3$. We can estimate these distributions with repeated sampling of consistent contrasts $\mathcal{D}_1$, $\mathcal{D}_2$, and $\mathcal{D}_3$, then averaging over sampling distributions to obtain sampling distributions $\hat{P}_1$, $\hat{P}_1$, and $\hat{P}_3$.

- $\hat{P}_1$: $\sum_{j \in 0,1} \sum_{b_{W=j}^{(i)}, d_{W=j}^{(i)}, v_{W=j}^{(i)}} \hat{P}\big(S_{W=1}, B_{W=1}, D_{W=1}, V_{W=1}, S_{W=0}, B_{W=0}, D_{W=0}, V_{W=0}\big)$
- $\hat{P}_2$: $\sum_{j \in 0,1} \sum_{w, d_{S=j}, v_{S=j}} \hat{P}(W, B_{S=1}, D_{S=1}, V_{S=1}, B_{S=0}, D_{S=0}, V_{S=0})$
- $\hat{P}_3$: $\sum_{j \in 0,1} \sum_{w, s, v_{B=j}} \hat{P}(W, S, D_{B=1}, V_{B=1}, D_{B=0}, V_{B=0})$

In each case $\hat{P}_i$ converges almost surely to $P_i$.

## F    MECHANICS IMPLEMENTED IN MULTIVERSE MECHANICA V1.0

In this section, we describe the ground truth causal structure and mechanics Multiverse Mechanica.

We describe each game mechanic in terms of:

- *level-2* interventional statements: A set of causal hypothetical statements of the form "Given preconditions $W$, all else equal, if $X$, then $Y$."
- A set of causal Markov kernels encompassing the relevant conditional probability distributions in the causal DAG.

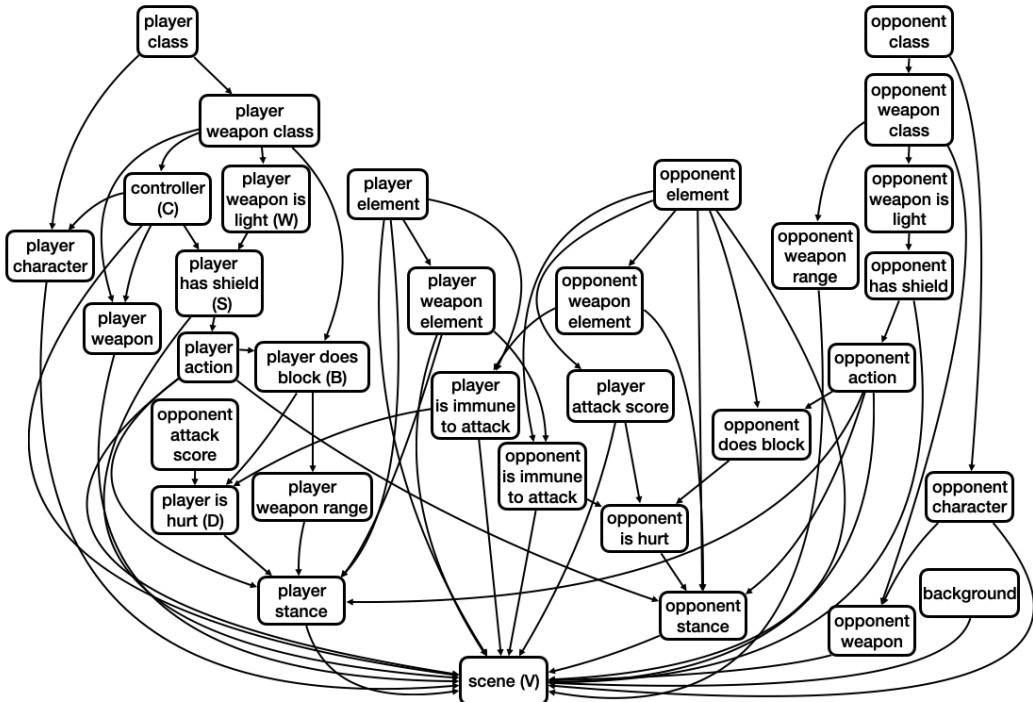

Figure 6: Full causal DAG of a turn in Multiverse Mechanica

- *level-3 parallel world* statements: A set of counterfactual statements of the form "Given preconditions $W$, all else equal, if $X$, then $Y$ AND if $X'$ then $Y'$."
- A set of counterfactual probability expressions defining the induced counterfactual outcome probabilities.

Particularly, in the case of the level-3 parallel world statements, we can enumerate counterfactual worlds in which we make a change and the target outcome variable(s) change, with guarantees on consistency, with respect to the game mechanic.

We formulate the counterfactual cases by taking interventions to change variables (**bold**), from the factual case (Case 0). Downstream changes are shown in *italics*.

**Full Causal DAG** Figure 6 shows the full causal DAG of a turn in Multiverse Mechanica v1.0.

## F.1 SHIELD MECHANIC

The warrior character can equip a shield they can use to block incoming attacks, if they have a free hand and perform the "defend" action.

### F.1.1 LEVEL-2 INTERVENTIONAL DEFINITION

We can completely define this mechanic by a set of contrasting statements:

- If a player has a small weapon, they may hold a shield, otherwise they cannot.
- If a player has a shield, they may block incoming attacks, otherwise they cannot.
- If a player blocks an incoming attack, they avoid taking damage, otherwise they may take damage.

**Involved variables:**

- `player action`: [idle, melee attack, defend]

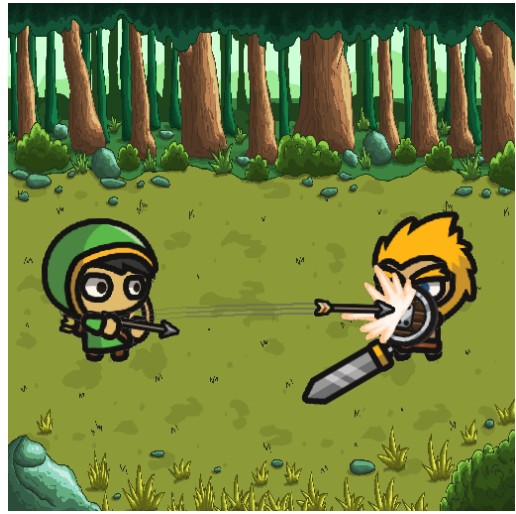 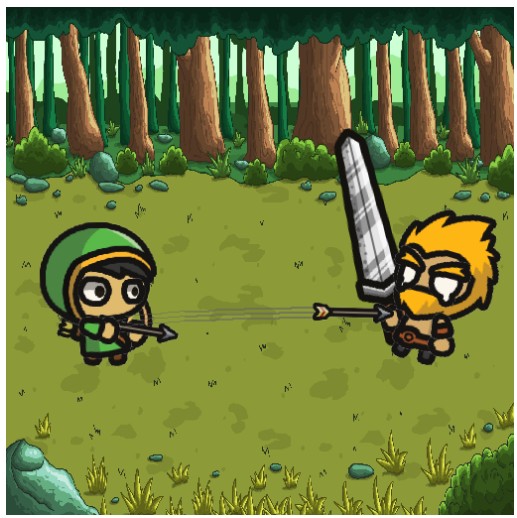

(a) Factual: The warrior is equipped with a one-handed short-sword and a shield, and blocks the arrow with a shield.

(b) Counterfactual: The warrior is equipped with a two-handed long-sword, and no shield, and is unable to block the attack.

Figure 7: Example counterfactual contrast statement for the shield mechanic.

- `opponent weapon class` : [small, large]
- `opponent has shield` : [True, False]
- `opponent action` : [idle, melee attack, defend]
- `opponent is hurt` : [True, False]

**Latent variables:**

- `opponent weapon is heavy` : [one-handed, two-handed]

### F.1.2 CAUSAL MARKOV KERNELS

In our model, *block* is one of the actions. Thus, we can reason about the shield game mechanic in terms of the player/opponent action being sampled as *block*.

The causal Markov kernel for the opponent action variable encompasses the shield game mechanic:

$$P\big(\text{opponent action} \mid \text{opponent has shield, opponent weapon}\big).$$

To reason about the opponent blocking, we consider the probability that the action variable is *block*:

$$P\big(\text{opponent action} = \text{block} \mid \text{opponent has shield, opponent weapon}\big).$$

To reason about the opponent getting hurt, we consider the probability that the opponent gets hurt, given the opponent action and player action:

$$P\big(\text{opponent is hurt} \mid \text{opponent action, opponent has shield, opponent weapon}\big).$$

An important detail to take note of is that the causal DAG does not have a variable to represent the character's outcome (e.g. getting hit, blocking, dodging)—it only has a notion of final stance. However, the stance variable is currently unused after sampling; the battle video game handles the underlying outcome logic and renders the scene.

### F.1.3 LEVEL-3 COUNTERFACTUAL PARALLEL WORLD STATEMENTS

We can rewrite the level-2 interventional statements as the following level-3 parallel world statements:

1. Given a character had a heavy weapon and they did not have a shield; if the character would have had a light weapon instead, then they could have equipped a shield.

2. Given a character had not had a shield and did not block; if the character had a shield instead, then they could have blocked.

3. Given a character had a shield but did not block and took damage; if the character had blocked instead, then they would have avoided taking damage from the attack.

### F.1.4 COUNTERFACTUAL PROBABILITY EXPRESSIONS

We can formulate these natural-language statements in mathematical notation.

**Notation.** Let:

- $WC$ = weapon class,
- $HS$ = has shield,
- $B$ = does block,
- $H$ = is hurt.

Then we have:

$$P\big(HS_{WC=\text{Light}} = \text{True} \,\big|\, WC = \text{Heavy},\ HS = \text{False}\big) > 0,$$
$$P\big(B_{HS=\text{True}} = \text{True} \,\big|\, HS = \text{False},\ B = \text{False}\big) > 0,$$
$$P\big(H_{B=\text{True}} = \text{False} \,\big|\, HS = \text{True},\ B = \text{False},\ H = \text{True}\big) = 1.$$

### F.2 ELEMENTAL IMMUNITY MECHANIC

Under the elemental immunity game mechanic, players may have elemental attributes; either fire or ice. Weapons may also have elemental attributes. If the attacking player weapon element and opponent element are the same (and non-none), the opponent is granted elemental immunity, and thus avoids taking damage from the attack, since they are already imbued with the element the attacker is using to attack them.

The elemental attributes of characters are indicated by variations from their base appearance:

- *none*: Characters appear in their standard appearances
- *fire*: Characters appear with orange shading and/or red outline
- *ice*: Characters appear with light blue shading and/or blue outline

The elemental attributes of weapons are similarly indicated by variations from their base appearance:

- *none*: Weapons appear in their standard appearances
- *fire*: Weapons appear with orange shading and/or red outline. Melee weapon slash visual effects appear with red shading. Ranged weapon projectiles (e.g., arrows, spell attacks) and explosions appear with red shading.
- *ice*: Weapons appear with light blue and/or blue outline. Melee weapon slash visual effects appear with blue shading. Ranged weapon projectiles and explosions appear with blue shading.

### F.2.1 LEVEL-2 INTERVENTIONAL DEFINITION

We can completely define this mechanic by a set of contrasting statements:

- If a player has an elemental attribute, they may wield a weapon with the same elemental attribute but they cannot wield one of an opposing element in the fire–ice dichotomy. For example, a player with a fire-elemental attribute may wield a fire-elemental weapon (or non-elemental weapon); they cannot wield an ice-elemental weapon.

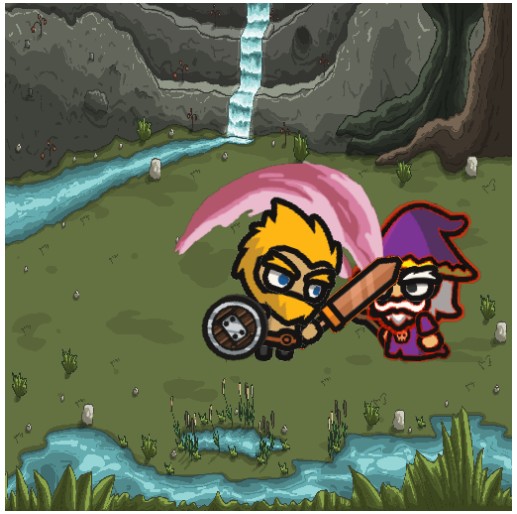 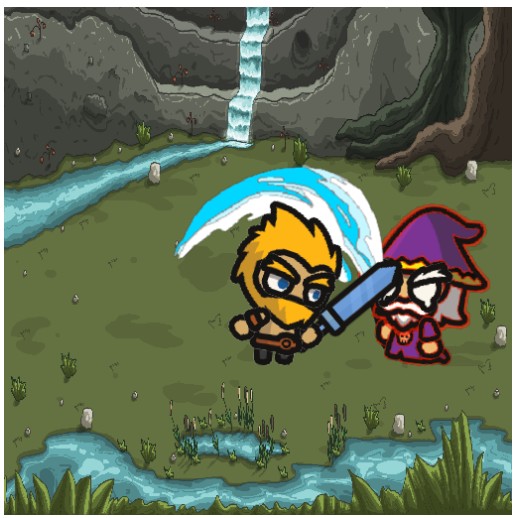

(a) Factual: The fire wizard is immune to the warrior's melee attack with a fire-type short sword.

(b) Counterfactual: The fire wizard takes damage from the warrior's melee attack with an ice-type short sword.

Figure 8: Example counterfactual contrast statement for the elemental immunity mechanic.

- If a player does not have an elemental attribute, they may wield any elemental-type weapon; otherwise the rule above applies.
- If a player is hit by an incoming attack from a weapon with the same elemental attribute, they are immune and thus avoid taking damage; otherwise they may take damage.

**Involved variables:**

- `player element` : [none, fire, ice]
- `player weapon element` : [none, fire, ice]
- `opponent element` : [none, fire, ice]
- `opponent is immune to attack` : [True, False]
- `opponent is hurt` : [True, False]

**Latent variables:** None.

### F.2.2 CAUSAL MARKOV KERNELS

The element of the player and the opponent are both independent variables:

$$P(\text{player element}),$$

$$P(\text{opponent element}).$$

The causal Markov kernel for the player weapon elemental is as follows:

$$P\big(\text{player weapon element} \mid \text{player element}\big).$$

To reason about the opponent being immune to the attack, we consider the conditional probability given the opponent's element and player's weapon element:

$$P\big(\text{opponent is immune to attack} \mid \text{player weapon element}, \text{opponent element}\big).$$

To reason about the opponent getting hurt, we consider the probability that the opponent gets hurt, given the opponent being immune:

$$P\big(\text{opponent is hurt} \mid \text{opponent is immune to attack}\big).$$

### F.2.3 LEVEL-3 COUNTERFACTUAL PARALLEL WORLD STATEMENTS

We can rewrite the level-2 interventional statements as the following level-3 parallel world statements:

1. Given a character had a fire elemental attribute and they had a fire elemental weapon; if the character would have had an ice elemental attribute instead, then they could have had an ice- or none-elemental weapon but not a fire elemental.

2. Given a character had a fire elemental attribute and they had a fire elemental weapon; if the character would have had no elemental attribute instead, then they could have had any elemental or non-elemental type weapon.

3. Given a character had an ice elemental attribute and was hit with an attack from an ice elemental weapon, and thus was immune and did not take damage; if the character had a fire elemental attribute instead, then they would have taken damage from the attack.

### F.2.4 COUNTERFACTUAL PROBABILITY EXPRESSIONS

We can formulate these natural language statements in mathematical notation.

**Notation.** Let:

- $PE$ = player element,
- $PWE$ = player weapon element,
- $OE$ = opponent element,
- $I$ = is immune to attack,
- $H$ = is hurt.

Then we have:

**Statement 1.**
$$P\big(PWE_{PE=\text{Ice}} = \text{Fire} \,\big|\, PE = \text{Fire}, \ PWE = \text{Fire}\big) = 0,$$
$$P\big(PWE_{PE=\text{Ice}} = \text{Ice} \,\big|\, PE = \text{Fire}, \ PWE = \text{Fire}\big) > 0,$$
$$P\big(PWE_{PE=\text{Ice}} = \text{None} \,\big|\, PE = \text{Fire}, \ PWE = \text{Fire}\big) > 0.$$

**Statement 2.**
$$P\big(PWE_{PE=\text{None}} = \text{Fire} \,\big|\, PE = \text{Fire}, \ PWE = \text{Fire}\big) > 0,$$
$$P\big(PWE_{PE=\text{None}} = \text{Ice} \,\big|\, PE = \text{Fire}, \ PWE = \text{Fire}\big) > 0,$$
$$P\big(PWE_{PE=\text{None}} = \text{None} \,\big|\, PE = \text{Fire}, \ PWE = \text{Fire}\big) > 0.$$

**Statement 3.**
$$P\big(H_{OE=\text{Fire}} = \text{True} \,\big|\, PWE = \text{Ice}, \ OE = \text{Ice}\big) = 1.$$

### F.3 WEAPON RANGE MECHANIC

Under the weapon range game mechanic, weapon classes have range attributes; either melee or ranged. Each weapon class is either a melee or ranged type. For example, the light sword, the heavy sword, and the dagger are melee weapons; while the bow, the staff, and the throwing knife are ranged weapons. The weapon range attribute affects primarily visual elements of the battle scene:

1. **Stance:** a character attacking with a melee weapon will be depicted in the snapshot physically swinging the melee weapon at the opponent's location, while a character attacking with a ranged weapon will be depicted shooting/throwing/casting from their position.

2. **Scene:** a character attacking with a melee weapon will be shown in the game scene (video) first approaching the opponent's position before physically swinging the melee weapon, while a character attacking with a ranged weapon will be depicted shooting/throwing/-casting from their position and the emitted projectile will be shown traveling towards the opponent from left to right (possibly on a parabolic trajectory, if affected by gravity, e.g. the arrow shot from the bow).

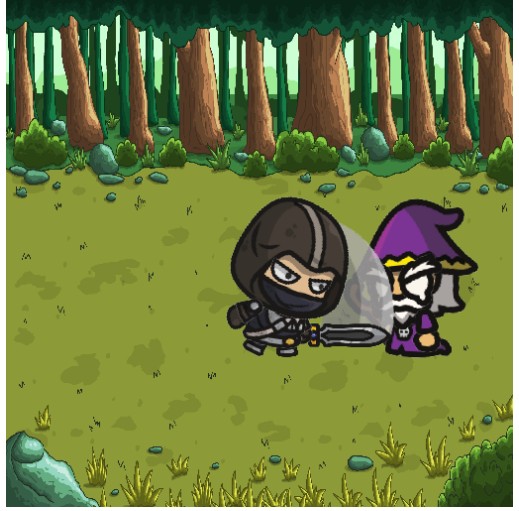 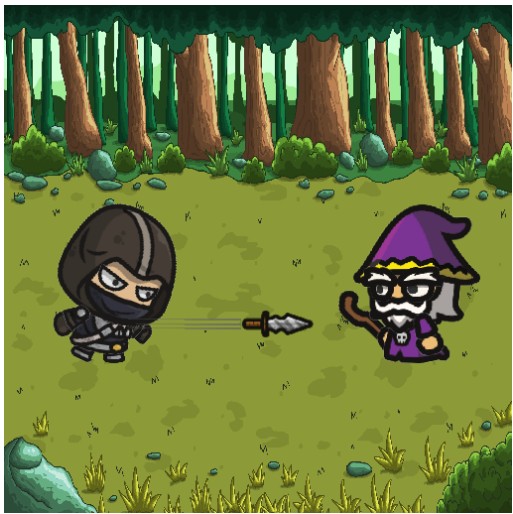

(a) Factual: The assassin performs a melee attack on a wizard with a sword.

(b) Counterfactual: The assassin performs a ranged attack on a wizard with a throwing knife.

Figure 9: Example counterfactual contrast statement for the weapon range mechanic.

### F.3.1 LEVEL-2 INTERVENTIONAL DEFINITION

We can completely define this mechanic by a set of contrasting statements:

- If a weapon is one of the weapon classes *light sword*, *heavy sword*, or *dagger*, it is a melee weapon; otherwise if it is *bow*, *staff*, or *throwing knife* it is a ranged weapon.
- If a player wields a melee weapon, they will be depicted in the snapshot as performing a melee attack at the opponent's location; otherwise they will be depicted as performing a ranged attack from their own location.
- If a player wields a melee weapon, they will be shown in the game scene (video) first approaching the opponent's position before physically swinging the melee weapon; otherwise they will be depicted shooting/throwing/casting from their position and the emitted projectile will be shown traveling towards the opponent from left to right.

**Involved variables:**

- `player weapon class` : [light sword, heavy sword, dagger, bow, staff, throwing knife]
- `player weapon range` : [melee, ranged]
- `player stance` : pixels in the rendered image snapshot
- `scene` : pixels in the rendered gameplay video

**Latent variables:** None.

### F.3.2 CAUSAL MARKOV KERNELS

The causal Markov kernel for the player weapon range is as follows:

$$P\big(\text{player weapon range} \mid \text{player weapon class}\big).$$

To reason about the player stance rendered in the image snapshot, we consider the conditional probability given the player weapon and player weapon range:

$$P\big(\text{stance} \mid \text{player weapon, player weapon range}\big).$$

Similarly, to reason about the scene rendered in the gameplay video, we consider the conditional probability given the player weapon, player weapon range, and player stance:

$$P\big(\text{scene} \mid \text{player weapon, player weapon range, stance}\big).$$

### F.3.3  LEVEL-3 COUNTERFACTUAL PARALLEL WORLD STATEMENTS

We can rewrite the level-2 interventional statements as the following level-3 parallel world statements:

1. Given a character had a light sword and they had a melee weapon; if the character would have had a bow instead, then they would have had a ranged weapon.

2. Given a character had a melee weapon and they were depicted in the snapshot as performing a melee attack at the opponent's location; if the character would have had a ranged weapon instead, then they would have been depicted as performing a ranged attack from their own location.

3. Given a character had a melee weapon and they were shown in the game scene video first approaching the opponent's position before physically swinging the melee weapon; if the character would have had a ranged weapon instead, then they would have been depicted shooting/throwing/casting from their position and the emitted projectile would have been shown traveling towards the opponent from left to right.

### F.3.4  COUNTERFACTUAL PROBABILITY EXPRESSIONS

We can formulate these natural language statements in mathematical notation.

**Notation.**  Let:

- $PWC$ = player weapon class,
- $PWR$ = player weapon range,
- $PS$ = player stance,
- $S$ = scene.

Then we have:

**Statement 1.**

$$P\big(PWR_{PWC=\text{bow}} = \text{ranged} \mid PWC = \text{light sword}, \ PWR = \text{melee}\big) = 1.$$

**Statement 2.**

$$P\Big(PS_{PWR=\text{ranged}} = \text{depicted performing ranged attack} \mid$$
$$PWR = \text{melee}, \ PS = \text{depicted performing melee attack}\Big) = 1, \tag{1}$$

$$P\Big(PS_{PWR=\text{ranged}} = \text{depicted performing melee attack} \mid$$
$$PWR = \text{melee}, \ PS = \text{depicted performing melee attack}\Big) = 0. \tag{2}$$

**Statement 3.**

$$P\Big(S_{PWR=\text{ranged}} = \text{shown performing ranged attack} \;\Big|$$

$$PWR = \text{melee}, \; PS = \text{shown approaching the opponent and performing melee attack}\Big) = 1,$$

$$(3)$$

$$P\Big(S_{PWR=\text{ranged}} = \text{shown performing melee attack} \;\Big|$$

$$PWR = \text{melee}, \; PS = \text{shown approaching the opponent and performing melee attack}\Big) = 0.$$

$$(4)$$

## F.4 SPELL-CASTING MECHANIC

Under the spell-casting mechanic, the wizard character may perform one of five spells:

1. **Spawn Magic Projectile Spell to Perform Ranged Attack**
2. **Summon Cloud Platform Spell to Dodge Attack**
3. **Self-Transform Spell to Increase Melee Strength**
4. **Opponent Transform Spell to Lower Enemy Defense**
5. **Levitation Spell to Disarm Opponent**

None of these spell actions can be mitigated by the enemy. Even in the case of the self-transform spell, the sheer size of the transformed wizard (into a golem) renders any block action by the opponent useless. Each spell action choice affects the *stance* and *scene* variables in the rendered game.

### F.4.1 LEVEL-2 INTERVENTIONAL DEFINITION.

All spells share the same base level-2 contrasting statements:

- If the player character is a wizard, they may wield a magical staff, otherwise they cannot.
- If the character is equipped with a magical staff, they may cast spells, otherwise they cannot.
- If the character casts a spell, they gain a particular offensive or defensive benefit to help them in the battle, otherwise they do not.

### F.4.2 CAUSAL MARKOV KERNELS

**Notation.** Let:

- $PC$ = player character
- $PW$ = player weapon
- $PA$ = player action
- $PF$ = player form
- $PS$ = player stance
- $OC$ = opponent character
- $OW$ = opponent weapon
- $OA$ = opponent action (e.g., defend)
- $OF$ = opponent form
- $OS$ = opponent stance
- $B$ = opponent successfully blocks (semantic variable, only true if the defend action actually succeeds)
- $D$ = opponent dodges

- $H$ = opponent is hurt
- Stance and Scene are the rendered variables

**Latent variables:**

- $OIA$ = opponent incoming attack indicator

**Shared enabling kernels.**

$$P(PW = \text{staff} \mid PC = \text{wizard}) > 0$$

$$P(PA \in \{\text{spell actions}\} \mid PW \neq \text{staff}) = 0$$

$$P(PA \mid PW = \text{staff, state}) \text{ is exogenous (agent policy)}$$

**Shared state & rendering kernels.**

$$P(PF, PS, OF, OS \mid PA, OA, PC, PW, OC, OW)$$

$$P(\text{Stance} \mid PF, PS, OF, OS, PA, OA)$$

$$P(\text{Scene} \mid \text{Stance}, PF, PS, OF, OS, PA, OA)$$

**Shared interaction/outcome kernels.**

$$P(B \mid PF, OA)$$

$$P(D \mid PS, OIA)$$

$$P(H \mid B, D, PA)$$

### F.4.3 LEVEL-3 COUNTERFACTUAL PARALLEL WORLD STATEMENTS.

All spells also share a common set of counterfactual statements:

1. Given a character was not a wizard and therefore could not wield a magical staff; if the character had been a wizard instead, then they could have wielded a magical staff.

2. Given a character was not equipped with a magical staff and therefore could not cast spells; if the character had been equipped with a staff instead, then they could have cast spells.

3. Given a character did not cast a spell and therefore received no benefit in the battle; if the character had cast a spell instead, then they would have gained the corresponding offensive or defensive benefit.

### F.4.4 COUNTERFACTUAL PROBABILITY EXPRESSIONS (SHARED).

*Not wizard → Wizard ⇒ Staff access*

$$P\big(PW_{PC=\text{wizard}} = \text{staff} \,\big|\, PC \neq \text{wizard}, PW \neq \text{staff}\big) > 0.$$

*No staff → Staff ⇒ Spell casting possible*

$$P\big(PA_{PW=\text{staff}} \in \{\text{spell actions}\} \,\big|\, PW \neq \text{staff}, PA \notin \{\text{spell actions}\}\big) > 0.$$

### F.5 SPAWN MAGIC PROJECTILE SPELL TO PERFORM RANGED ATTACK

The wizard can cast a spell to conjure and launch a magical projectile directly at the opponent. This action is a ranged attack, and its visual and mechanical dynamics follow the same principles as other ranged weapons described in Section F.3.

### F.5.1 LEVEL-2 INTERVENTIONAL DEFINITION.

- If the wizard casts the spawn projectile spell, a magical projectile is launched towards the opponent (i.e., $OIA =$ True), otherwise no projectile is spawned ($OIA =$ False).
- If a magical projectile strikes the opponent and is not successfully blocked or dodged, the opponent is hurt, otherwise they are not.

**Involved variables:**

- $PA =$ player action
- $OA =$ opponent action (e.g., defend, dodge)
- $PS =$ player stance
- $B =$ opponent successfully blocks
- $D =$ opponent dodges
- $H =$ opponent is hurt

**Latent variables:**

- $OIA =$ opponent incoming attack indicator

### F.5.2 CAUSAL MARKOV KERNELS.

$$P(OIA \mid PA)$$
$$P(B \mid PF, OA)$$
$$P(D \mid PS, OIA)$$
$$P(H \mid B, D, PA)$$

### F.5.3 LEVEL-3 COUNTERFACTUAL PARALLEL WORLD STATEMENTS.

1. Given the wizard idled and there was no incoming attack; if the wizard had cast the spawn projectile spell instead, then the opponent would have faced an incoming attack ($OIA =$ True).

2. Given the wizard cast the spawn projectile spell and the projectile hit (i.e., no successful block or dodge) and the opponent was hurt; if the opponent had successfully blocked or dodged instead, then they would not have been hurt.

### F.5.4 COUNTERFACTUAL PROBABILITY EXPRESSIONS.

*Idle $\rightarrow$ Projectile spell $\Rightarrow$ Incoming attack present*
$$P\big(OIA_{PA=\text{projectile spell}} = \text{True} \,\big|\, PA = \text{idle}, OIA = \text{False}\big) = 1.$$

*Projectile hits $\rightarrow$ Successful block $\Rightarrow$ Not hurt*
$$P\big(H_{B=\text{True}} = \text{False} \,\big|\, PA = \text{projectile spell}, OIA = \text{True}, B = \text{False}, H = \text{True}\big) = 1.$$

*Projectile hits $\rightarrow$ Successful dodge $\Rightarrow$ Not hurt*
$$P\big(H_{D=\text{True}} = \text{False} \,\big|\, PA = \text{projectile spell}, OIA = \text{True}, D = \text{False}, H = \text{True}\big) = 1.$$

### F.5.5 SUMMON CLOUD PLATFORM SPELL TO DODGE ATTACK

The wizard has a spell they can use to summon a levitating cloud platform and use it to raise themselves up above the battlefield to dodge incoming enemy attacks.

**level-2 Interventional Definition.**

- If the wizard casts the cloud platform spell, a platform is summoned; the wizard moves onto it and is levitated upwards above the battlefield, otherwise they remain on the ground.
- If the wizard is in an elevated position on a platform during an incoming attack, they successfully dodge and avoid damage, otherwise if they are on the ground they remain vulnerable and may take damage.

**Involved variables:**

- $PA$ = player action
- $OA$ = opponent action
- $OS$ = opponent stance (grounded, elevated)
- $H$ = opponent is hurt

**Latent variables:** None.

**Causal Markov Kernels.**

$$P(OS \mid PA)$$
$$P(D \mid OS, OIA),$$
$$P(H \mid D).$$

**level-3 Counterfactual Parallel World Statements.**

1. Given the wizard performed the idle action and remained on the ground; if the wizard had cast the cloud platform spell instead, then they would have been elevated.

2. Given the wizard remained on the ground during an incoming attack and was hurt; if the wizard had cast the cloud platform spell instead, then they would have been elevated and dodged the attack, and thus not been hurt.

**Counterfactual Probability Expressions.**

*Idle $\rightarrow$ Cloud Platform $\Rightarrow$ Elevated stance*

$$P\big(OS_{PA=\text{cloud platform}} = \text{elevated} \,\big|\, PA = \text{idle}, OS = \text{grounded}\big) = 1.$$

*Grounded, hurt during incoming attack $\rightarrow$ Cloud Platform $\Rightarrow$ Dodge, not hurt*

$$P\big(H_{OS=\text{elevated}} = \text{False} \,\big|\, PA = \text{idle}, OS = \text{grounded}, OIA = \text{True}, H = \text{True}\big) = 1.$$

### F.5.6 SELF-TRANSFORM SPELL TO INCREASE MELEE STRENGTH

The wizard can cast a spell to transform themselves into a large golem. In this form, their melee attacks are enormously strengthened and cannot be blocked by opponents.

**level-2 Interventional Definition.**

- If the wizard casts the self-transform spell, they transform into a golem, otherwise they remain in their normal form.
- If the wizard is transformed into a golem, their melee attack cannot be blocked due to sheer size and strength, otherwise it can.
- If the wizard's golem-form melee attack is not successfully blocked, the opponent will be hurt, otherwise they will not be.

**Involved variables:**

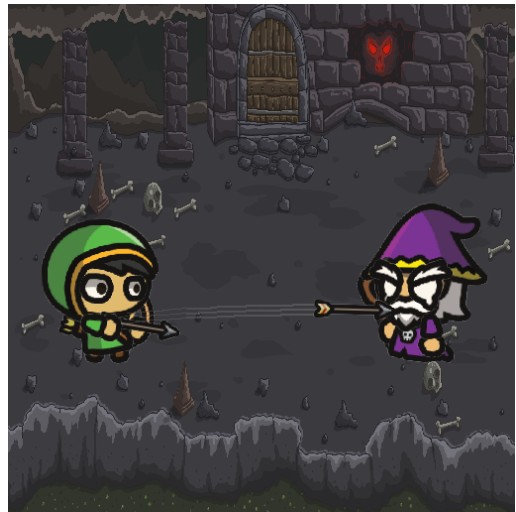 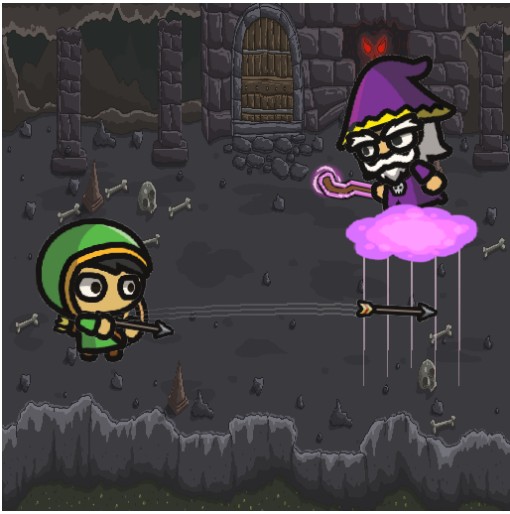

(a) Factual: The wizard idles and is struck by the archer's arrow.

(b) Counterfactual: The wizard summons a magical cloud platform to gain an elevated position and dodge the archer's arrow.

Figure 10: Example counterfactual contrast statement for the summon cloud platform spell mechanic.

- $PA$ = player action
- $OA$ = opponent action (e.g., defend)
- $B$ = opponent successfully blocks
- $H$ = opponent is hurt

**Latent variables:**

- $PF$ = player form (normal, golem)

**Causal Markov Kernels.**
$$P(PF \mid PA)$$
$$P(B \mid PF, OA)$$
$$P(H \mid B, PA)$$

**level-3 Counterfactual Parallel World Statements.**

1. Given the wizard performed the idle action and their form remained unchanged; if the wizard had cast the self-transform spell instead, then they would have transformed into a mighty golem.
2. Given the wizard cast a projectile spell and it was blocked by the opponent; if the wizard had cast the self-transform spell instead, then they would have transformed into a golem and the melee attack would not have been blocked by the opponent.
3. Given the wizard cast a projectile spell and it was blocked by the opponent and thus they were not hurt; if the wizard had cast the self-transform spell instead, then they would have transformed into a golem and the melee attack would have bypassed the block and hurt the opponent.

**Counterfactual Probability Expressions.**

*Idle → Self-Transform ⇒ Golem form*
$$P\big(PF_{PA=\text{self-transform}} = \text{golem} \,\big|\, PA = \text{idle}, PF = \text{normal}\big) = 1.$$

*Magic projectile attack blocked → Self-Transform ⇒ Block attempt failed*

$$P\big(B_{PF=\text{golem}} = \text{True} \,\big|\, PA = \text{magic projectile attack}, PF = \text{normal}, OA = \text{defend}, B = \text{True}\big) = 0.$$

*Magic projectile attack blocked, not hurt → Self-Transform ⇒ Hurt despite block action*

$$P\big(H_{PF=\text{golem}} = \text{True} \,\big|\, PA = \text{magic projectile attack}, PF = \text{normal}, OA = \text{defend}, H = \text{False}\big) = 1.$$

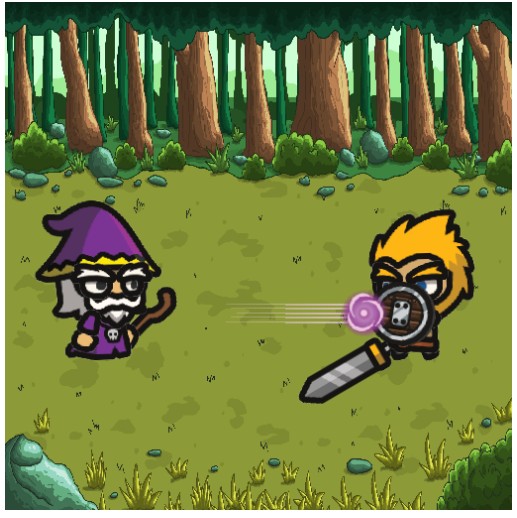 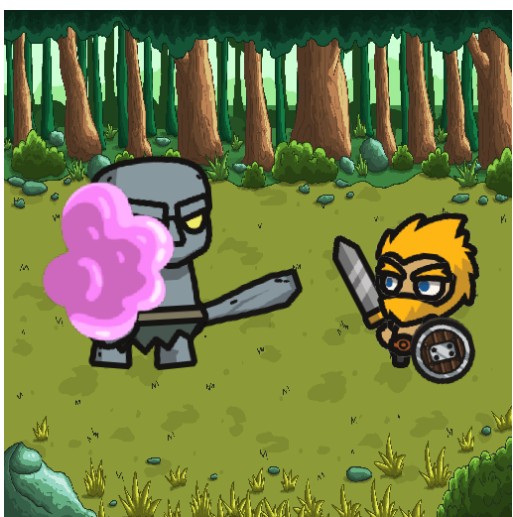

(a) Factual: The wizard casts a magic projectile spell but the warrior blocks it.

(b) Counterfactual: The wizard casts a spell to transform itself into a large golem to perform a powerful melee attack.

Figure 11: Example counterfactual contrast statement for the self-transform spell mechanic.

### F.5.7 OPPONENT TRANSFORM SPELL TO LOWER ENEMY DEFENSE

The wizard can cast a spell to transform an opponent into a weak or harmless creature (i.e., a snail), disarming them and preventing them from defending against incoming attacks.

**level-2 Interventional Definition.**

- If the wizard casts no spell, the opponent retains their normal form and can block or defend as usual.
- If the wizard casts the opponent transform spell, the opponent is transformed (e.g. into a snail), otherwise they remain in their normal form.
- If the opponent is transformed, they are disarmed and cannot block; otherwise they may block.

**Involved variables:**

- $PA$ = player action
- $OA$ = opponent action
- $B$ = opponent successfully blocks
- $H$ = opponent is hurt

**Latent variables:**

- $OF$ = opponent form (normal, transformed)

**Causal Markov Kernels.**

$$P(OF \mid PA),$$
$$P(H \mid OF, B).$$

**level-3 Counterfactual Parallel World Statements.**

1. Given the wizard cast a projectile spell and the opponent remained in their normal form; if the wizard had cast the opponent transform spell instead, then the opponent would have been transformed into a snail.

2. Given the wizard cast a projectile spell and it was blocked by the opponent, and thus the opponent was not hurt; if the wizard had cast the opponent transform spell instead, then the opponent would have been unable to block and would have been hurt by a follow-up attack.

**Counterfactual Probability Expressions.**

*Idle or projectile spell → Opponent Transform ⇒ Opponent form changes*

$$P\big(OF_{PA=\text{opponent transform}} = \text{transformed} \,\big|\, PA \neq \text{opponent transform}, OF = \text{normal}\big) = 1.$$

*Projectile spell blocked, not hurt → Opponent Transform ⇒ Block disabled (opponent disarmed)*

$$P\big(B_{OF=\text{transformed}} = \text{True} \,\big|\, PA = \text{magic projectile attack}, OF = \text{normal}, B = \text{True}\big) = 0.$$

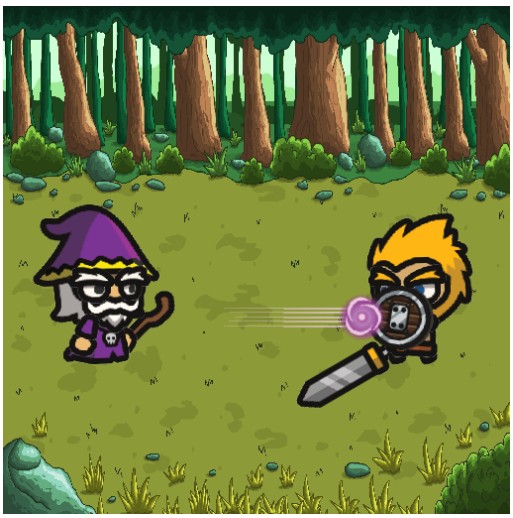
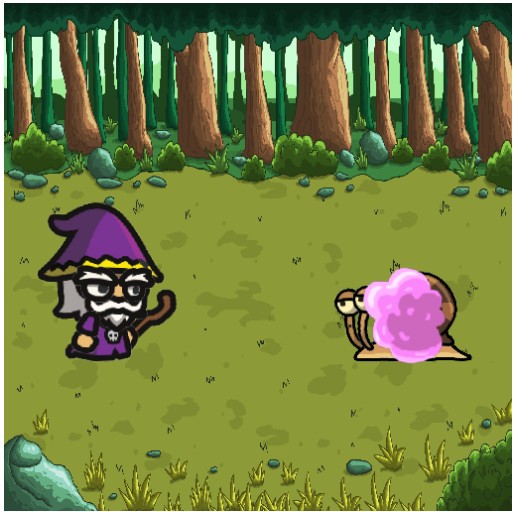

(a) Factual: The wizard casts a magic projectile spell but the warrior blocks it.

(b) Counterfactual: The wizard casts a spell to transform the warrior into a snail to disarm them.

Figure 12: Example counterfactual contrast statement for the opponent transform spell mechanic.

### F.5.8 LEVITATION SPELL TO DISARM OPPONENT

The wizard can cast a spell that lifts the opponent into the air, leaving them unable to defend or block effectively, and rendering them disarmed.

**level-2 Interventional Definition.**

- If the wizard casts no spell, the opponent remains grounded and may block normally.
- If the wizard casts the levitation spell, the opponent is lifted into the air, otherwise they remain grounded.

- If the opponent is levitated, they cannot block and are effectively disarmed; otherwise they may block.

**Involved variables:**

- $PA$ = player action
- $OA$ = opponent action
- $OS$ = opponent stance (grounded, levitating)
- $B$ = opponent successfully blocks
- $H$ = opponent is hurt

**Latent variables:** None.

**Causal Markov Kernels.**

$$P(OS \mid PA),$$
$$P(H \mid OS, B)$$

**level-3 Counterfactual Parallel World Statements.**

1. Given the wizard cast a projectile spell and the opponent remained grounded; if the wizard had cast the levitation spell instead, then the opponent would have been levitated.

2. Given the wizard cast a projectile spell and it was blocked by the opponent, and thus the opponent was not hurt; if the wizard had cast the levitation spell instead, then the opponent would have been unable to block and left vulnerable.

**Counterfactual Probability Expressions.**

*Idle or projectile spell → Levitation ⇒ Opponent stance changes*

$$P\big(OS_{PA=\text{levitation}} = \text{levitating} \,\big|\, PA \neq \text{levitation}, OS = \text{grounded}\big) = 1.$$

*Projectile spell blocked, not hurt → Levitation ⇒ Block disabled (opponent disarmed)*

$$P\big(B_{OS=\text{levitating}} = \text{True} \,\big|\, PA = \text{magic projectile attack}, OS = \text{grounded}, B = \text{True}\big) = 0.$$

## G   GAME DESIGN DECISIONS

To ensure that the datasets produced by our simulation respect the causal assumptions of our model, we designed the game architecture with consistency guarantees as a primary objective. This appendix documents how these guarantees were realized in practice, focusing on two complementary aspects: (i) the modular, system-based design of game mechanics that mirrors the structure of the causal model, and (ii) the temporal alignment of captured frames to ensure semantic consistency in dynamic interactions.

### G.1   BRIDGING GAME MECHANICS AND CAUSAL MECHANISMS WITH SYSTEM-BASED DESIGN

At the foundation of our data generation pipeline lies a design principle: game mechanics must be implemented in a way that respects and preserves the structure of the causal model they instantiate. To achieve this, we adopted a fully modular *Entity–Component–System* (ECS) architecture (Nystrom, 2014; Gregory, 2018), which enforces locality of causal mechanisms and supports reproducibility across runs.

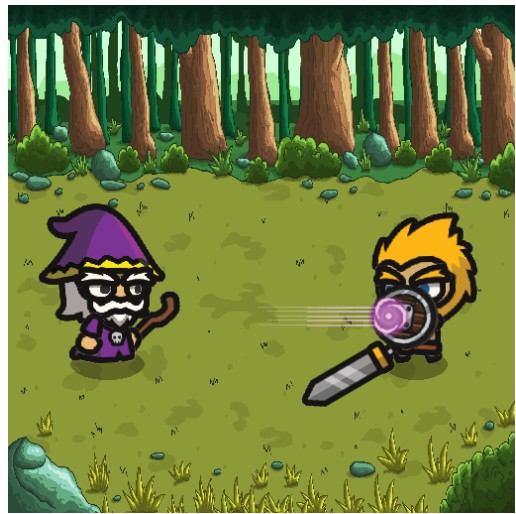
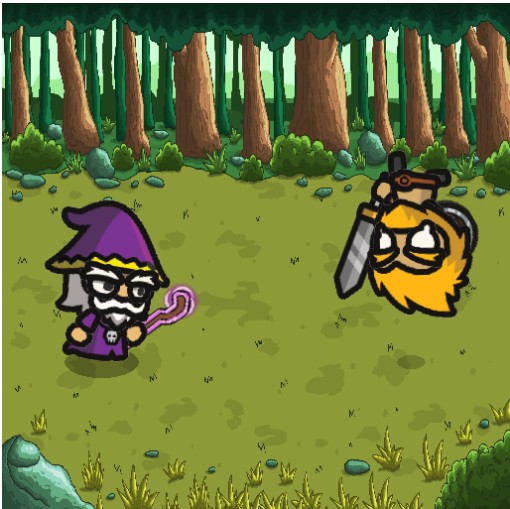

(a) Factual: The wizard casts a magic projectile spell but the warrior blocks it.

(b) Counterfactual: The wizard casts a levitation spell to lift the warrior off the ground, disarming them.

Figure 13: Example counterfactual contrast statement for the levitation spell mechanic.

### G.1.1 ENTITIES, COMPONENTS, AND SYSTEMS AS CAUSAL UNITS.

In our implementation, *entities* represent the units of analysis (e.g., characters, projectiles, platforms), *components* represent their attributes and state (e.g., position, animation phase, shield presence), and *systems* encapsulate the transformation rules that govern state evolution (e.g., combat resolution, kinematics, physics, or screenshot scheduling). This separation guarantees that each causal mechanism is expressed locally, without entanglement with unrelated processes.

Each system implements a distinct causal mechanism: for example, the `GameCombatSystem` maps action states of attacker and defender into outcomes such as *hit*, *block*, or *immune*, while the `GamePhysicsSystem` governs the motion of projectiles according to deterministic physical rules. Systems are generally designed to operate frame-by-frame using component data as inputs, but they may also maintain local state when required (for example, the `GameAnimationSystem` manages a per-entity priority queue of animation requests). This design ensures that each causal transformation is encapsulated and modular, while still supporting the persistent state needed for realistic simulation.

### G.1.2 ALIGNMENT WITH CAUSAL GRAPH STRUCTURE

The ECS architecture was chosen deliberately to reflect the modular structure of our causal model. Nodes in the causal graph correspond to entity attributes (e.g., *weapon class*, *stance*, *elemental immunity*), while edges correspond to the update dependencies realized through system logic.

For example, the transition of a defender into a *blocking* or *hurt* state depends on multiple upstream components: the presence of a shield component, the character's action selection component (indicating whether the chosen action is to block), and downstream trigger flags such as `needs_to_block` or `is_hurt`. The shield and action selection components establish the potential for blocking, while the trigger flags are set based on situational context (e.g., proximity of an incoming attack). Only when both preconditions and triggers align does the `GameCombatSystem` update the defender's state to *blocking*; otherwise, the state transitions to *hurt*.

This design directly encodes the causal mechanism:

shield component + action selection $\longrightarrow$ {`needs_to_block`, `is_hurt`} $\longrightarrow$ outcome (block or hurt).

By structuring dependencies in this way, the system preserves the logic of the causal graph within the mechanics of the game engine. This mapping ensures that system update rules correspond closely to the assignment functions of the causal model.

### G.1.3 LOCALITY AND MODULARITY FOR CONSISTENCY

By localizing mechanics in dedicated systems, the architecture prevents hidden confounding across game features. For instance, animation timing is managed exclusively by the `GameAnimationSystem`, while collision and trajectory updates are confined to the `GamePhysicsSystem`. This guarantees that modifying one mechanism (e.g., projectile gravity) does not inadvertently alter another (e.g., collision detection or blocking). Such modularity enforces a form of "causal isolation," allowing dataset generation to reflect the true structure of the designed model.

### G.1.4 REPRODUCIBILITY AND CONNECTION TO INTERVENTIONS

The ECS structure also guarantees reproducibility: since each system applies deterministic update rules to the current component state, identical initial conditions yield identical traces. Importantly, the systems themselves do not support interventions in the sense of directly overriding assignment functions during simulation. Instead, interventions are handled at the model level: sampled values from the causal model are passed into the simulation as inputs that determine which branches of the `GameBehaviorTree` are executed (e.g., sampled values specifying whether a character *does block*). The behavior tree then orchestrates the scene by triggering the appropriate system updates, while each system executes its assignment function deterministically given the requested state changes. In this way, the game engine acts as a faithful executor of causal mechanisms, while the intervention logic is confined to the sampling layer above.

Through this design, the simulation environment operates as a direct computational analogue of the causal model, where each mechanism is encapsulated in a corresponding system. This guarantees that generated training data inherits the same modularity and independence properties as the underlying causal graph, thereby supporting consistency-guaranteed counterfactual analysis.

### G.2 IMPACT FRAMES: DEFINING SEMANTIC CONSISTENCY IN DYNAMIC CAUSAL MODEL TRACES VIA *Point of Maximum Action* CONCEPT

While the system-based architecture guarantees local causal consistency at the level of game logic, temporal alignment must also be addressed to preserve counterfactual consistency in dynamic interactions. To this end, our system generates gameplay clips of contrasting player turns, each designed to include a canonical *impact frame*: the instant an attack connects, a shield block occurs, or a projectile visibly misses. Ensuring that these per-turn impact frames correspond to semantically equivalent points in the causal process is critical; otherwise, contrasts risk reflecting phase misalignment rather than true causal differences. We therefore formalize alignment using the *Point of Maximum Action* (PoMA) principle, which anchors impact frames to the most visually and mechanically expressive phase of the interaction.

### G.2.1 CONCEPTUAL FRAMING: TEMPORAL ALIGNMENT FOR COUNTERFACTUAL COMPARISONS

In static structural causal models (SCMs), counterfactuals are evaluated at a single time index, and semantic alignment across factual and counterfactual worlds is immediate. In dynamic SCMs, the meaning of an event depends on *when* it occurs relative to the unfolding process. For example, a melee strike might connect later than a projectile impact due to differences in action duration. If frames are extracted at fixed indices, we risk capturing non-equivalent phases of these interactions (e.g., an attack wind-up in one turn versus a point of contact in another). If we extract training artifacts (e.g., impact frames) at fixed indices, we risk capturing non-equivalent phases of these interactions (such as a wind-up in one run versus a point of contact in another). This undermines the validity of counterfactual comparisons by introducing differences that are artifacts of temporal phasing rather than consequences of the intervention.

We therefore treat each player turn as a temporal *causal trace* and align counterfactual observations to the *most informative temporal locus* of the relevant event class. We formalize this with the principle of the *Point of Maximum Action* (PoMA).

### G.2.2   POINT OF MAXIMUM ACTION (POMA).

Let $A(S_{t'}) \in \mathbb{R}_{\geq 0}$ score how *action-intense* the counterfactual state $S_{t'}$ is with respect to the target event class (e.g., impact, block). The PoMA alignment selects

$$t'_{\text{PoMA}} \;=\; \arg\max_{t' \in T'} A(S_{t'}).$$

PoMA frames are then extracted at $t'_{\text{PoMA}}$, aligning the impact frame to the point of maximum expressivity. This guarantees that contrasts correspond to the same semantic phase of the interaction, regardless of variation in action duration.

### G.2.3   FOUR ALIGNMENT METHODS: BRIEF SUMMARY WITH PROS AND CONS

We summarize four practical approaches for temporal alignment of dynamic counterfactuals. Each provides a distinct trade-off between simplicity, robustness, and semantic fidelity.

**1) Constant Time Interval.** *Rule.* Evaluate the counterfactual variable at the same nominal time as the factual: $t' \equiv t$. *Pros.* Conceptually simple; trivial to implement; deterministic. *Cons.* Fails when action durations differ; risks capturing non-equivalent phases (e.g., mid-swing vs. impact); unsuitable for dynamic interactions where event timing adapts to interventions.

**2) Equilibrium-Based.** *Rule.* Evaluate once the counterfactual dynamics have reached a steady state or absorbing condition (e.g., $\|S'_{t'+1} - S'_{t'}\| < \varepsilon$ or a domain predicate holds). *Pros.* Appropriate for tasks where long-run properties matter; robust to transient phasing differences; alignment invariant to small shifts in sequence length. *Cons.* Inapplicable to inherently transient events (e.g., impacts); some episodes may not converge; equilibrium may erase the very distinctions needed to analyze acute causal effects.

**3) Point of Maximum Action (PoMA).** *Rule.* Align to the counterfactual time of peak action intensity for the event class:

$$t'_{\text{PoMA}} \;=\; \arg\max_{t'} A(S'_{t'}).$$

*Pros.* Directly targets the most salient phase of the interaction; robust to differences in sequence length; naturally accommodates variable-duration actions by abstracting to their peak. *Cons.* Requires a well-defined intensity score $A$; can be non-trivial for abstract or multi-agent interactions; may require smoothing when peaks are brief or noisy.

**4) Semantic Consistency.** *Rule.* Align by maximizing semantic similarity between factual and counterfactual states. *Pros.* General and flexible in principle; useful in settings where semantic descriptors are available. *Cons.* Not used in our implementation. It requires an additional similarity metric and embedding design, which introduces complexity and potential bias.

In practice, we adopt the PoMA approach, extending it with event-specific scoring functions and event-defined windows to handle variable-duration interactions. Constant Time and Equilibrium serve primarily as conceptual baselines, while semantic similarity was considered but not implemented.

### G.2.4   IMPLEMENTATION IN OUR GAME: EVENT-BASED WINDOWS AND WEIGHTED SCORING

To instantiate these principles in a reproducible data pipeline, our engine implements an event-driven *GameScreenshotSystem* that schedules captures at semantically aligned moments:

1. **Event Detection.** The simulation raises structured events for interactions of interest (e.g., `melee_impact`, `projectile_hit`, `shield_block`). Each event is associated with the participating entities and their current states.

2. **Event-Based Scoring Windows.** For each interaction type, we define a start event and a stop event that bound a scoring window (e.g., `swing_start` → `impact`, or `projectile_cast` → `collision`). These windows are managed directly in the behavior tree, ensuring that scoring only occurs during the semantically relevant phase of the interaction.

3. **Weighted Scoring.** Within each window, frames are scored according to event-specific weights. For example, projectile impact events may be given higher weight than projectile flight, and shield contact may be prioritized over shield raise. The capture frame is then chosen as

$$\widehat{t'} \in \underset{t' \in \text{window}}{\arg\max} A(S'_{t'}),$$

with deterministic tie-breaking for reproducibility.

By anchoring artifact capture to event-defined scoring windows and applying weighted intensity scoring, our pipeline produces semantically aligned visual data across simulations, despite natural variability in action durations. This guarantees that counterfactual comparisons reflect genuine causal differences, rather than artifacts of capturing frames at arbitrary, non-equivalent time indices.

### G.3 SUMMARY

Taken together, these design choices preserve consistency at both structural and temporal levels. The ECS architecture ensures local mechanics map cleanly to modular causal mechanisms, while the screenshot system anchors each player turn contrast to a canonical *impact frame* selected via PoMA. By aligning contrasts at the most expressive phase of interaction, the system guarantees that observed differences reflect genuine causal effects rather than timing artifacts, producing impact frames that are both causally and semantically consistent.

## H SPECIFICATIONS OF GENERATED VIDEO CLIPS

Generated video clips have following technical specifications:

### H.1 RESOLUTION AND FORMAT

All video clips are rendered at a resolution of $512 \times 512$ pixels in a square aspect ratio. Videos are encoded in MP4 format using the H.264 codec with yuv420p pixel format to ensure broad compatibility across video players and analysis frameworks.

### H.2 FRAME RATE, DURATION, AND TIMING

Videos are captured at 50 frames per second (FPS) using a fixed-interval delta time method to ensure consistent temporal sampling across all generated clips. This approach decouples game simulation time from wall-clock time, enabling reproducible frame timing essential for dataset generation and comparative analysis.

Video clip duration is variable and event-driven, spanning the complete battle sequence from initialization to completion. Clip length is determined by the termination of both player behavior trees, typically ranging from several seconds to longer sequences depending on the complexity of actions performed (e.g., melee attacks, spell casting, projectile interactions, defensive maneuvers).

### H.3 DATASET ORGANIZATION

The system generates consistent contrasts, enabling direct comparison between observed battle outcomes and alternative scenarios under modified conditions. Each video file follows the naming convention `battle_XXXXXX-TIMESTAMP.mp4`, where `XXXXXX` represents a zero-padded 6-digit battle identifier. Accompanying JSON metadata files provide technical details including encoding parameters, frame timing information, and battle configuration data.

### H.4 IMPLEMENTATION

Video generation utilizes the imageio library with PyAV backend for efficient encoding. The rendering pipeline captures frame sequences from the game's screenshot buffer system, which also maintains temporal consistency in impact frames through the behavior tree execution framework.

## H.5 VARIABLES GENERATED

In addition to clips, each generated example includes:

- Controller inputs $C_t$
- Full or partial state $X_t$
- Mechanic-specific CL-DAG $G^{\mathcal{M}}$
- Set of mechanic-specific parallel world DAGs
- Set of mechanic-specific counterfactual DAGs
- Seed / $\omega$ identifiers

# I SUGGESTED METRICS

| Task | Description | Metric |
|------|-------------|--------|
| Fully observed mechanic inference | Infer $\mathbf{P_i}$ using clips, game state variables, and controller inputs | $D_{\mathrm{KL}}(P_i \,\|\, \hat{P}_i)$ |
| Partial (canonical) mechanic inference | Infer $\mathbf{P_i}$ using clips and controller inputs, assuming game state is unobserved | $D_{\mathrm{KL}}(P_i \,\|\, \hat{P}_i)$ |
| Generate consistent contrasts | Generate multiple consistent contrast examples for each $d_i$ associated with the mechanic, and validate that they are visually accurate and consistent | Human or VLM validation |
| Counterfactual abduction | Generate $\{v_{X=x}, v_{X=x'}\}$ from game, obtain $\hat{v}_{X=x'} = \mathbb{E}[V_{X=x'} \mid V_{X=x} = v_{X=x}]$ using the model, then compare $\hat{v}_{X=x'}$ to $v_{X=x'}$ | Human or VLM validation |

Table 5: Tasks that can be used to evaluate mechanic learning with Multiverse Mechanica.

Table 5 summarizes the four tasks for evaluating mechanic-learning with *Multiverse Mechanica*. We expand here on how each metric is operationalized and its relation to prior work.

## I.1 MECHANIC INFERENCE.

The fully- and partially-observed inference tasks require estimating the distributions $\hat{P}_i$ implied by mechanic-specific constraints $\mathcal{M}$ and comparing them against the ground truth distributions $P_i$. We adopt KL-divergence as the primary quantitative measure, explicitly reporting $D_{\mathrm{KL}}(P_i \,\|\, \hat{P}_i)$ for each mechanic. This aligns with classical practice in distributional evaluation, where divergence to the ground truth provides a direct measure of whether the model has captured the stochastic relations induced by the mechanic.

## I.2 CONSISTENCY IN CONTRAST GENERATION.

The contrast-generation and counterfactual-abduction tasks require evaluating whether pairs of clips $(v_{X=x}, v_{X=x'})$ are *consistent*, i.e. differing only in outcomes attributable to the intervention variable $X$ while holding non-descendant factors fixed. Prior work has typically relied on human evaluation: for example, Gingerson et al. (2024) collected large-scale human judgments of whether generated gameplay videos adhered to intended mechanics. Recent work has investigated whether VLMs can serve as automated evaluators of model generations. Hendriksen et al. (2025) show that pretrained VLMs can be adapted for evaluating world model rollouts, e.g. scoring whether predicted videos match textual descriptions of target outcomes. However, to our knowledge, no prior work has specifically used VLMs to evaluate *consistency* across parallel-world contrasts, as defined by causal counterfactual principles. This remains an open direction, and *Multiverse Mechanica* provides level-3 parallel-world data where such evaluations can be systematically explored.

Work by Monteiro et al. (2023) introduced *composition* metrics that effectively evaluate consistency in image-based counterfactuals by checking whether attributes remain unchanged under controlled edits. These metrics capture aspects of counterfactual faithfulness that parallel our notion of consistency. However, they have not yet been extended to video data, nor applied to generative modeling of game mechanics. We view *Multiverse Mechanica* as a platform to develop such extensions, allowing both human and VLM-based evaluation methods to be compared side-by-side.

### I.3 SUMMARY.

In short, our metrics combine classical distributional divergences for inference tasks with human- or VLM-based consistency checks for contrastive generation tasks. This dual approach ensures that models are evaluated not only on reproducing distributions of state variables but also on capturing the causal consistency of gameplay dynamics under controlled interventions.

## J THEORETICAL AND IMPLEMENTATION DETAILS FOR PROOF-OF-CONCEPT

### J.1 BACKGROUND ON DIFFUSION MODELS AND REVERSE-SAMPLING

Denoising diffusion probabilistic models (DDPMs) define a forward Markov chain that gradually adds Gaussian noise to an image and a learned reverse process that denoises it step by step. Given a data sample $x_0 \sim q(x_0)$, the forward process produces $x_t$ by directly adding noise to the data: $q(x_t|x_{t-1}) = \mathcal{N}\big(x_t; \sqrt{\alpha_t}x_{t-1}, (1 - \alpha_t)I\big)$ for $t = 1, \ldots, T$. Here $0 < \alpha_t < 1$ are predefined variances. In the reverse generation, one starts from pure noise $x_T \sim \mathcal{N}(0, I)$ and iteratively denoises to $x_0$ using a model $\epsilon_\theta$ trained to predict the injected noise. At each step, the model predicts $\hat{\epsilon} \approx \epsilon$ such that $x_{t-1}$ can be estimated by removing noise: e.g., $x_{t-1} = \frac{1}{\sqrt{\alpha_t}}\big(x_t - (1 - \alpha_t)\hat{\epsilon}\big)$ (with additional variance for stochastic sampling). In practice, one can also use the continuous-time formulation and solve a reverse stochastic differential equation or its deterministic counterpart (as in DDIM), yielding a mapping from an initial noise $u$ directly to an image $x = G(u, c)$. Importantly, this exogenous noise $u$ acts as the stochastic latent that accounts for random variation in generated images. Using a deterministic sampler (e.g.,setting $\eta = 0$ in DDIM), one obtains a one-to-one mapping between $u$ and the output $x$, and can invert a given image to its corresponding $u$ for a particular conditioning $c$.

We will leverage this invertibility to extract the latent noise $u_a$ from the original image $x$ and the latent noise $u_b$ from the counterfactual image $x_{\text{cf}}$, where $x$ denotes the factual image generated under the original conditioning, $x_{\text{cf}}$ represents the counterfactual image generated under modified conditioning, $u_a$ is the inverted latent noise corresponding to the factual image $x$, and $u_b$ is the inverted latent noise corresponding to the counterfactual image $x_{\text{cf}}$.

### J.2 BACKGROUND ON CAUSAL COUNTERFACTUALS IN IMAGE GENERATION

In causal terms, we can view the generative model as a structural causal model $x := f_\theta(u, c)$, where $c$ is a cause (e.g.,textual description or a set of discrete random variables) and $u$ is an unobserved exogenous variable accounting for randomness. A counterfactual image aims to answer: "What would the image look like if we change $c$ from $c_A$ to $c_B$, while keeping all other latent factors the same?" The classical procedure for generating such counterfactuals is the three-step abduction-action-prediction: (1) Abduction: infer the exogenous noise $u_a$ that produced the factual image $x$ under $c_A$ (this $u_a$ captures the instance-specific variations of $x$); (2) Action: intervene by setting the prompt to $c_B$ (while keeping $u_a$ fixed); (3) Prediction: generate the new image as $x_{\text{cf}} = f_\theta(u_a, c_B)$. This procedure, if the model perfectly captures the true causal mechanism, would change only the aspects of the image directly affected by $c$ and leave other details intact (satisfying causal consistency that no non-descendant features of $c$ change). In practice, directly using $u_a$ with a new prompt $c_B$ can produce a reasonable edited image, but it may fail or produce artifacts if $c_B$ demands alterations that conflict with the original latent factors.

By contrast, many non-SCM image editing approaches do not explicitly enforce the same latent noise. For example, one might simply prompt the model with $c_B$ and generate a new sample (different $u$), or apply heuristics like latent interpolation, attention refocusing, or mask-based noising of only certain regions. These approaches can produce plausible results, but often lack guarantees that only the intended changes occur—the model might inadvertently change unrelated details because

the random draw $u$ or other generation conditions differ. Our goal is to incorporate causal principles into the diffusion editing process to maximize counterfactual faithfulness (only $c$-dependent changes) while still allowing the model flexibility to implement the edit realistically.

## J.3 CONTRASTIVE TRAINING VIA ALIGNMENT LOSSES

In this section, we provide the necessary background to help understand our method.

## J.4 NOTATION

We first define the notation used in our counterfactual editing framework. Let $x$ denote the original (factual) image, and let $x_{\mathrm{cf}}$ denote the counterfactual image we aim to generate (the edited image after an intervention). We model image generation via a diffusion model as $x = G(u, c)$, where $u$ is an initial exogenous noise (drawn from a Gaussian prior, typically $u \sim \mathcal{N}(0, I)$) and $c$ is the conditioning (in our case, a text prompt). We use $c_A$ for the original prompt and $c_B$ for the counterfactual prompt. Using an inversion technique (e.g., reverse ODE or deterministic DDIM inversion), we can obtain $u_a$ as the noise that generates $x$ under $c_A$, and similarly $u_b$ as the noise corresponding to $x_{\mathrm{cf}}$ under $c_B$. We denote by $x_t$ the (noisy) latent image at diffusion timestep $t$ when evolving toward $x$ (with $x_0 = x$ and $x_T = u_a$ for the forward noising process), and likewise $x_{\mathrm{cf},t}$ for the counterfactual trajectory. The diffusion model's denoiser is denoted $\epsilon_\theta(x_t, c, t)$, which predicts the added noise at step $t$ for latent $x_t$ and conditioning $c$. For brevity, we write $\epsilon(x_t, c, t)$ when $\theta$ is clear from context. Finally, $\mathcal{L}_1$, $\mathcal{L}_2$, $\mathcal{L}_{\mathrm{text}}$, and $\mathcal{L}_{\mathrm{sub}}$ will denote different loss terms introduced below.

## J.5 METHOD 1: $L_1$ – CONSISTENCY ALIGNMENT

Our first new loss function enforces consistency alignment between the factual and counterfactual generations. We obtain the noise $u_a$ and $u_b$ corresponding to $x$ and $x_{\mathrm{cf}}$ respectively (via the inversion process described above). The consistency alignment loss is then defined as,

$$\mathcal{L}_1 = \|u_a - u_b\|_2^2, \quad u_* = H_\theta^{T \leftarrow 0}(x_*, c_*) \tag{5}$$

where $H_\theta^{T \leftarrow 0}$ represents the inversion function that maps from image space back to noise space at timestep $T$.

Given the diffusion model $x = f_\theta(u, c)$, we have:

$$x = f_\theta(u_a, c_A) \tag{6}$$
$$x_{\mathrm{cf}} = f_\theta(u_b, c_B) \tag{7}$$

The inversion takes the image back to the noise, which yields:

$$u_a = H_\theta^{T \leftarrow 0}(x, c_A) \tag{8}$$
$$u_b = H_\theta^{T \leftarrow 0}(x_{\mathrm{cf}}, c_B) \tag{9}$$

The $\mathcal{L}_1$ is added to the training loss as a regularization term to enforce ***exogenous invariance***. In the ideal case where $u_b = u_a$, the counterfactual generation becomes:

$$x_{\mathrm{cf}} = f_\theta(u_a, c_B) \tag{10}$$

which ensures that all variations between $x$ and $x_{\mathrm{cf}}$ are attributed solely to the conditioning change $c_A \rightarrow c_B$, while preserving the exogenous factors encoded in $u_a$.

This loss directly penalizes any differences between the underlying noise vectors of the original and edited image. The motivation is to ensure that $x$ and $x_{\mathrm{cf}}$ share the same source of variation, so that as much of the scene's random details as possible remain unchanged. This approach extracts editing-related information from the seed, enabling differences to be more expressed by the conditioning $c$ rather than random variations. The primary change affects the reverse mapping/generator's decomposition of conditions, belonging to "seed-level" invariance.

Intuitively, if $u_a$ and $u_b$ are identical, the only differences between $x_{\mathrm{cf}}$ and $x$ will come from the changed conditioning $c_B$ vs $c_A$. In the ideal case, $\mathcal{L}_1 = 0$ means we are generating the counterfactual with the exact same "random seed" as the factual image (the pure SCM counterfactual). This

encourages maximal consistency: the background, lighting, style, and other incidental attributes should stay the same unless the new prompt explicitly demands their change. Enforcing $\mathcal{L}_1$ provides strong alignment that leads to stable edits. It prevents the edited image from drifting in appearance or composition: the counterfactual will tend to have the same objects and layout as the original, only differing in the aspects dictated by the prompt change. This is beneficial for preserving identity (e.g.,the same person's face before and after an edit) and ensuring only the intended attributes change.

However, this strict constraint can also have a few limitations. If the counterfactual prompt $c_B$ is significantly different from $c_A$, using exactly the same noise $u_a$ might overly constrain the generation, resulting in artifacts or an incomplete edit. The model might struggle to reconcile $u_a$ (which was optimal for the original content) with the new prompt, leading to implausible images or failure to fully achieve the desired change.

### J.6 Method 2: $L_2$ – Structure Preservation at High-Noise

As a more relaxed alternative, we propose to align the diffusion model's behavior for the two images at the high-noise stages of generation, rather than forcing the initial noises to be identical. Concretely, let $S$ be a set of diffusion time steps focusing on the high-noise region (e.g.,the latter half or last third of the diffusion schedule, when $x_t$ is still highly noisy). We define the structure preservation loss $\mathcal{L}_2$ as:

$$\mathcal{L}_2 = \sum_{t \in S} \|\epsilon_\theta(x_t, c_A, t) - \epsilon_\theta(x_{\text{cf},t}, c_B, t)\|_2. \tag{11}$$

$\mathcal{L}_2$ measures the disparity between the model's denoising predictions for the factual versus counterfactual image trajectories, but only at very noisy states (where $x_t$ is mostly noise). By penalizing this difference, we encourage the denoiser's reaction to the two inputs to be the same in the early stages of generation. This effectively steers $x_{\text{cf},t}$ to evolve in a similar direction as $x_t$ while the image is still coarse and noisy, ensuring the two generation processes start out aligned in terms of global structure. Importantly, $L_2$ does not enforce that the latent noises $x_t$ themselves are exactly equal, only that the predicted noise residuals (or equivalently, the score vectors) are similar. This distinction makes $L_2$ a partial relaxation of the $L_1$ constraint. It nudges the counterfactual to have a similar high-level appearance without locking in all the exact stochastic details.

When using $L_2$, the model is free to adjust $u_b$ as needed, but it will still preserve large-scale aspects of $u_a$. For example, if $x$ depicts a particular scene layout, $L_2$ will bias $x_{\text{cf}}$ to keep that layout, since early denoising steps (which shape the overall composition) will be similar for both. As $t$ gets smaller (less noise), $x_{\text{cf}}$ can gradually diverge more to realize the new content $c_B$ specifies. This approach maintains structure and identity better than an unconstrained edit, while granting more flexibility than $L_1$ for the model to incorporate the new prompt. In essence, $L_2$ focuses on aligning the coarse-grained features (which are determined in high-noise stages) and lets the fine details emerge freely.

One might consider applying a spatial mask so that structure preservation is enforced only on certain regions (for instance, only aligning the background areas that should remain unchanged). In our approach, we generally did not require an explicit mask for $L_2$. Since $L_2$ operates on high-noise (low-detail) states, it inherently affects global structure more than specific fine features. We found that a well-balanced $L_2$ encourages overall consistency without needing per-pixel restrictions—the model naturally preserves unedited parts of the image. However, if a particular application demands strict locality (e.g., editing only a small region while leaving everything else exactly as is), a mask could be introduced to further ensure no influence of $L_2$ on the region to be changed (or conversely, to focus $L_2$ only on the region to preserve). In summary, $L_2$ already provides a soft, global consistency constraint, and masking is an optional refinement rather than a necessity in most cases.

### J.7 Abduction–Action–Prediction and Its Diffusion Emulation

#### J.7.1 SCM Setup.

Let $\mathcal{M} = (\mathbf{U}, \mathbf{V}, \mathbf{F}, P(\mathbf{U}))$ be a structural causal model with exogenous variables $\mathbf{U}$, endogenous variables $\mathbf{V}$, structural assignments $\mathbf{F}$, and exogenous distribution $P(\mathbf{U})$ (Pearl, 2009). We single

out: (i) the *mechanic variables* $X \subseteq \mathbf{V}$ that we will intervene on; (ii) the *controller input* $C$; and (iii) the visual variable $V$ whose realization is a impact frame snapshot $v$. Throughout, we explicitly treat our generation seed $\omega$ as a realization of the SCM exogenous variables, i.e., $\omega \sim P(\mathbf{U})$, and we treat all other shared rendering conditions as part of $\omega$ (while $C$ is held fixed explicitly).

### J.7.2 ABDUCTION–ACTION–PREDICTION (AAP).

Given a factual observation $v_0$ obtained under $(X{=}x_0, C{=}c)$, the AAP recipe for the two-world case proceeds as:

**(Abduction)** Infer $P(\mathbf{U} \mid V{=}v_0, X{=}x_0, C{=}c)$, choose a representative $\hat{\omega}$.

**(Action)** Form the intervened model $\mathcal{M}_{X=x_1}$ while keeping $C{=}c$ and $\mathbf{U}{=}\hat{\omega}$ fixed.

**(Prediction)** Evaluate the counterfactual $V_{X=x_1}(\hat{\omega})$.

Equivalently, in distributional form,

$$\hat{v}_1 \sim P\big(V_1 \,\big|\, V_0{=}v_0,\, X_0{=}x_0,\, C{=}c\big),$$

which is shorthand for propagating $\hat{\omega} \sim P(\mathbf{U} \mid V_0{=}v_0, X_0{=}x_0, C{=}c)$ through $\mathcal{M}_{X=x_1}$.

---

**Algorithm 1** Abduction–Action–Prediction (two-world case)

---

**Require:** SCM $\mathcal{M} = (\mathbf{U}, \mathbf{V}, \mathbf{F}, P(\mathbf{U}))$, factual $(v_0, x_0, c)$, target $x_1$
1: **Abduction:** Infer $\hat{\omega} \leftarrow$ MAP/mean/sample from $P(\mathbf{U} \mid V{=}v_0, X{=}x_0, C{=}c)$
2: **Action:** Construct $\mathcal{M}_{X=x_1}$; hold $C{=}c$ and $\mathbf{U}{:=}\hat{\omega}$
3: **Prediction:** Compute $\hat{v}_1 \leftarrow V_{X=x_1}(\hat{\omega})$
4: **return** $\hat{v}_1$

---

### J.8 DIFFUSION-BASED EMULATION OF AAP

In our latent diffusion setting, we emulate abduction–action–prediction (AAP) by identifying the SCM exogenous variables with the model's initial latent noise:

$$\mathbf{U} \longleftrightarrow \omega \sim \mathcal{N}(0, I).$$

**Abduction (DDIM inversion).** We estimate $\hat{\omega}$ from a factual impact frame $V_{X=x_0}$ under $(x_0, c)$ via deterministic sampler inversion (DDIM, $\eta{=}0$). Let $Z_{t,X=x_0}$ denote its noisy latents across $t \in [0, T]$. At each step, the denoiser $\epsilon_\theta(Z_{t,X=x_0}, t, c_0)$ predicts the noise, and inversion equations (Appendix J.10.5) are used to propagate forward in the schedule, yielding an estimate $\hat{\omega} = \mathsf{Abduct}_\theta(Z_{0,X=x_0}, c_0)$.

**Action.** Hold $c$ fixed and change the mechanic state from $x_0$ to $x_1$.

**Prediction (deterministic reverse).** Initialize the trajectory with $Z_{T,X=x_1} := \hat{\omega}$ and run the deterministic reverse process conditioned on $(x_1, c)$ to obtain a counterfactual latent $Z_{0,X=x_1}$, then decode it to the counterfactual impact frame $\hat{V}_{X=x_1}$.

---

**Algorithm 2** Diffusion-based AAP via DDIM ($\eta{=}0$)

---

**Require:** Diffusion model ($\epsilon_\theta$, scheduler, decoder), factual $(V_{X=x_0}, x_0, c)$, target $x_1$
1: **Abduction:** $\hat{\omega} \leftarrow \mathsf{Abduct}_\theta(Z_{0,X=x_0}, c_0)$          // DDIM inversion in latent space
2: **Action:** Keep $c$ fixed, set $X := x_1$
3: **Prediction:** $Z_{T,X=x_1} := \hat{\omega}$; for $t = T, \ldots, 1$: $Z_{t-1,X=x_1} \leftarrow \mathsf{DDIMStep}\big(Z_{t,X=x_1}, t, x_1, c; \epsilon_\theta\big)$;
      $\hat{V}_{X=x_1} \leftarrow \mathsf{decoder}(Z_{0,X=x_1})$
4: **return** $\hat{V}_{X=x_1}$

---

### J.8.1 CONNECTION TO OUR TRAINING LOSSES.

The AAP framing clarifies the roles of our loss components. (i) *Exogenous alignment* $\mathcal{L}_1$ encourages shared-seed invariance by driving

$$\mathsf{Abduct}_\theta(Z_{0,X=x_0}, c_0) \approx \mathsf{Abduct}_\theta(Z_{0,X=x_1}, c_1),$$

for elements of the same contrast that share the true $\omega$. This ensures the abduction step produces consistent seeds across worlds. (ii) *Structure preservation* $\mathcal{L}_2$ aligns denoiser outputs $\epsilon_\theta(Z_{t,X=x_j}, t, c_j)$ at high-noise steps $t \in S$, enforcing agreement on coarse global structure during the early reverse process. This mirrors AAP's assumption that exogenous factors ($\omega$) are held fixed while only $X$ changes.

### J.8.2 SCM $\leftrightarrow$ Diffusion mapping (two-world case).

| SCM concept | | Diffusion instantiation |
|---|---|---|
| Exogenous $\mathbf{U}$ | $\leftrightarrow$ | Initial latent noise seed $\omega \sim \mathcal{N}(0, I)$ |
| Abduction $P(\mathbf{U} \mid V_{X=x_0}, x_0, c)$ | $\leftrightarrow$ | DDIM ($\eta=0$) inverse sampling $\hat{\omega} = \mathsf{Abduct}_\theta(Z_{0,X=x_0}, c_0)$ |
| Action $X = x_1$ | $\leftrightarrow$ | Condition sampling process on $(x_1, c)$ |
| Prediction $V_{X=x_1}(\hat{\omega})$ | $\leftrightarrow$ | Deterministic sampling to $Z_{0,X=x_1}$ then decode $\hat{V}_{X=x_1}$ |
| Causal consistency | $\leftrightarrow$ | Shared seed $\hat{\omega}$; early-step structure preservation ($\mathcal{L}_2$) |

Abduction via DDIM inversion yields an *estimate* $\hat{\omega}$ whose fidelity depends on the schedule and conditioning; see Appendix J.10.6 for caveats and tuning guidance.

### J.9 Additional Regularizers

Beyond the core losses $L_1$ and $L_2$, our framework can incorporate additional terms to improve consistency and fidelity:

### J.9.1 Subspace Consistency Loss.

We can encourage the factual and counterfactual images to remain close in certain intermediate representations of the diffusion model. For example, one may align hidden latents or cross-attention maps at corresponding diffusion steps. By penalizing differences in these subspaces (e.g., the model's multi-head attention maps for background tokens, or feature maps in a particular UNet layer), we enforce that the internal generation pathways for $x$ and $x_{\text{cf}}$ stay similar. This helps preserve layout and identity at a semantic level, complementing the pixel-space alignment enforced by $L_1/L_2$. Formally, if $F_t(x)$ denotes some feature (such as a latent embedding or attention tensor) computed during the denoising of $x$ at step $t$, we can define a loss $L_{\text{sub}} = \sum_{t \in \mathcal{T}} \|F_t(x) - F_t(x_{\text{cf}})\|_2$ for some chosen set of layers or timesteps $\mathcal{T}$. This subspace consistency loss encourages the edited image to differ only minimally in features unrelated to the intervention.

**Text Consistency Loss.** To ensure the edited image indeed reflects the counterfactual prompt $c_B$, we include a text-image consistency term. We rely on a pretrained image-text similarity model (such as CLIP) to measure alignment between $x_{\text{cf}}$ and the description $c_B$. Let $\text{sim}(x_{\text{cf}}, c_B)$ be a similarity score (higher means the image matches the prompt better). We define a loss $L_{\text{text}} = -\text{sim}(x_{\text{cf}}, c_B)$ (or equivalently $1 - \text{sim}$, depending on the normalization) so that minimizing $L_{\text{text}}$ maximizes the agreement between the generated image and the desired attributes. This ensures that while preserving content, we do not under-shoot the edit: the new image should clearly exhibit the prompted change. The text consistency loss guides the generation to remain faithful to the user's request, especially when $L_1$ or $L_2$ are pulling towards the original image. It helps avoid the outcome where the edit is so conservative that the difference between $x_{\text{cf}}$ and $x$ is hard to discern.

### J.10 Loss Combination

Our full counterfactual editing objective combines these components in a weighted sum:

$$L_{\text{total}} = \lambda_1 L_1 + \lambda_2 L_2 + \lambda_3 L_{\text{text}} + \lambda_4 L_{\text{sub}}, \tag{12}$$

where $\lambda_i$ are tunable weights that control the influence of each loss term. In practice, we choose these weights to balance identity preservation against effective editing. Typical values and trade-offs are as follows:

$\lambda_1$ **(Consistency Alignment):** This is often kept relatively small (e.g., $\lambda_1$ in the range 0 to 0.5) unless the edit is very minor. A small $\lambda_1$ nudges the initial noise vectors closer without forcing identity completely. Increasing $\lambda_1$ leads to more literal counterfactuals (very high consistency with the original image's details), but if set too high it may prevent the new attributes from appearing strongly. There is a trade-off between maintaining background/identity (higher $\lambda_1$ favors this) and allowing change (lower $\lambda_1$ gives more freedom).

$\lambda_2$ **(Structure Preservation):** We usually give $L_2$ a moderate to high weight (on the order of 1.0) as it is the principal mechanism to preserve structure. $\lambda_2$ in a range roughly 0.5 to 2.0 works well. A larger $\lambda_2$ tightly constrains the high-level layout and style to match the original, which is good for identity preservation; however, if $\lambda_2$ is excessively large, it can act almost as strictly as $L_1$, potentially impeding necessary changes. Reducing $\lambda_2$ allows the counterfactual generation to deviate more in composition if needed, but too low $\lambda_2$ might result in unwanted alterations in background or other objects.

$\lambda_3$ **(Text Consistency):** This weight should be high enough to ensure the edit actually happens (especially for subtle changes), but not so high that it overrides the preservation losses. In practice $\lambda_3$ is often set around 0.5 to 1.0 (assuming similarity is scaled to a comparable range) so that the image aligns with the prompt without artifacts. If $\lambda_3$ is set too low, the edit might be too conservative (the model might simply regenerate the original image to satisfy $L_1/L_2$). If $\lambda_3$ is too high, the model may introduce exaggerated or incorrect features to satisfy the prompt, possibly compromising identity or visual quality.

$\lambda_4$ **(Subspace Consistency):** If used, this is typically a small auxiliary weight (e.g., 0.1). Since $L_{\text{sub}}$ operates on internal features, it can strongly bind the generation if overweighted. A modest $\lambda_4$ helps reinforce structural consistency without conflicting with the primary losses. Tuning $\lambda_4$ involves checking that it indeed improves preservation of details like face identity or scene layout, without, for example, freezing the image in an early-state that ignores the new prompt. In some cases, we might set $\lambda_4 = 0$ (i.e., not use this term) if we find $L_1$ and $L_2$ are sufficient; when used, it serves as an extra regularizer.

In summary, $L_1$ and $L_2$ provide a spectrum between strict and loose alignment, $\lambda_3$ drives the fidelity to the requested counterfactual change, and $\lambda_4$ can bolster consistency on a feature level. We recommend starting with a balanced combination (for instance, $\lambda_1 = 0.2, \lambda_2 = 1.0, \lambda_3 = 0.5, \lambda_4 = 0.1$).

### J.10.1 DIFFUSION BACKBONE.

We use a pretrained Stable Diffusion variant with a deterministic sampler [2], which makes the mapping between exogenous noise $u$ and image $v$ approximately invertible. This enables abduction of $u$ from an image and consistent reuse across parallel worlds. Classifier-free guidance is applied with scale 7.5, and the scheduler uses $T = 50$ steps.

### J.10.2 CONDITIONING.

In the main text we describe conditioning directly on game state variables (e.g., shield, weapon type, block outcome). For implementation, these variables and outcomes are converted into natural-language captions for compatibility with CLIP text encoders. This is a nuisance parameterization: the underlying conditioning remains the game state variables.

### J.10.3 TRAINING SETUP.

We fine-tune the network backbone (UNet) only, keeping the VAE and text encoder frozen. Batch size is 4, training runs for 50 epochs, with a cosine learning-rate schedule, weight decay 0.01, and gradient clipping. Images are peak-action snapshots extracted at canonical times in each episode to minimize temporal ambiguity.

### J.10.4 ALIGNMENT LOSSES.

We mainly use two loss functions for finetuning:

---

[2]specifically we use DDIM $\eta = 0$, also the deterministic samplers in Karras et al. (2022)

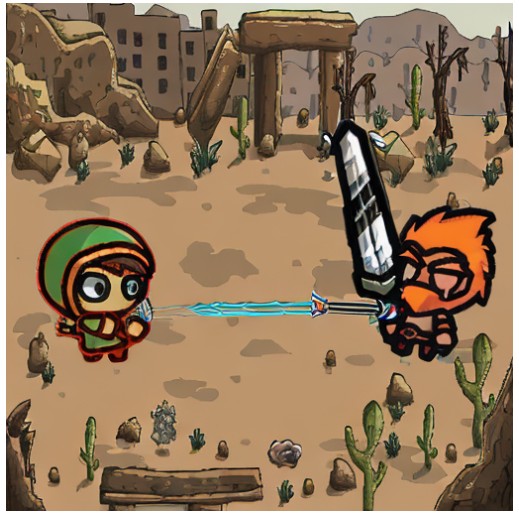 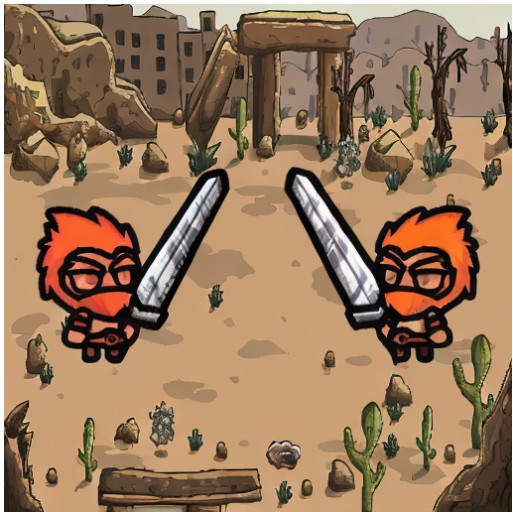

(a) Factual: A 1-on-1 battle. Foe 1 type is archer, element is fire, weapon is bow, weapon element is none, has shield is false, action is shoot. Foe 2 type is warrior, element is fire, weapon is long sword, weapon element is none, has shield is false, action is idle. Foe turn is foe 1.

(b) Counterfactual: A 1-on-1 battle. Foe 1 type is warrior, element is fire, weapon is long sword, weapon element is none, has shield is false, action is idle. Foe 2 type is warrior, element is fire, weapon is long sword, weapon element is none, has shield is false, action is idle. Foe turn is foe 1.

Figure 14: Example counterfactual image pairs generated from finetuned text-to-image diffusion model.

- $\mathcal{L}_1$: consistency alignment. After inverting both factual and counterfactual images to latent noise $(u_a, u_b)$, we penalize $\|u_a - u_b\|^2$, enforcing invariance of exogenous factors. Note that this loss function is very expensive due to it needs to inverse two sampling paths for each counterfactual data pair. In practice, we only apply $\mathcal{L}_1$ to one data point per batch of training data.
- $\mathcal{L}_2$: structure preservation at high noisy region (large SNRs). At early diffusion timesteps, we penalize discrepancies between denoiser predictions for factual vs. counterfactual trajectories, encouraging global structural consistency.

Both terms can be weighted with coefficients $\lambda_1, \lambda_2$.

### J.10.5 SEED ABDUCTION WITH DETERMINISTIC DIFFUSION.

Our multiverse alignment objective requires that consistent contrasts share the same exogenous noise $\omega$. In deterministic samplers (e.g., DDIM), this means that two parallel reverse processes $(Z_{t,X=x_0})_{t=0}^T$ and $(Z_{t,X=x_1})_{t=0}^T$ can be initialized with the same $\omega$, ensuring non-descendant content is consistent.

Given an observed impact frame $V_{X=x}$ with controller input $c$, let $Z_{0,X=x}$ denote its clean latent before decoding and $Z_{t,X=x}$ the noisy latent at step $t$. At each step, the denoiser $\epsilon_\theta(\cdot)$ predicts noise $\epsilon_\theta(Z_{t,X=x}, t, c)$, which is then used to trace the trajectory backward through the noise schedule. Iterating these updates recovers an estimate $\hat{\omega} = \mathsf{Abduct}_\theta(V_{X=x}, c)$.

Applying this inversion procedure to both members of a contrast yields $\hat{\omega}_{x_0}$ and $\hat{\omega}_{x_1}$, which ideally coincide. The seed-consistency loss $\mathcal{L}_1$ penalizes their distance, providing a concrete operationalization of the causal consistency principle within deterministic diffusion.

### J.10.6 CAVEATS.

**Deterministic inversion.** We use DDIM with $\eta = 0$ for $\mathsf{Abduct}_\theta$, yielding an approximately bijective mapping between seed and latent trajectory under fixed conditioning and schedule. In practice, invertibility is approximate and sensitive to: (i) the precise noise schedule; (ii) classifier-free guid-

ance settings; and (iii) conditioning $(x, c)$. Hence $\hat{\omega}$ should be treated as a consistent *estimate* rather than a ground-truth latent.

### J.10.7 CHOOSING THE HIGH-NOISE SET $S$ FOR $\mathcal{L}_2$.

We select $S$ as either (i) the last $k$ steps of the schedule (empirically $k \in [\frac{T}{3}, \frac{T}{2}]$), or (ii) all $t$ with $\text{SNR}(t) \leq \tau$ for a threshold $\tau$. A weighted variant uses $w_t \propto \text{SNR}(t)^{-\gamma}$ and

$$\mathcal{L}_2^{\text{w}} = \sum_t w_t \left\| \epsilon_\theta(z_{0,t}, t, x_0, c) - \epsilon_\theta(z_{1,t}, t, x_1, c) \right\|_2^2.$$

Under mild conditions, aligning score predictions at high-noise is connected to alignment in data space via Stein's identity.

### J.10.8 SCOPE.

This is a feasibility study. We claim no pixel-level counterfactual identification and provide only qualitative illustrations. Future work may extend this approach to video sequences and more complex mechanics.

## K SOFTWARE DEPENDENCIES

### K.1 GAME ENGINE.

The game itself is implemented in `Pygame`, a lightweight Python library for 2D graphics and interaction. We chose Pygame because it enables rapid prototyping of turn-based combat mechanics, frame-accurate rendering of impact frames, and reproducible control of random seeds, all within a Python environment that integrates smoothly with machine learning workflows.

### K.2 CAUSAL MODELING.

To formalize and simulate the causal generative process underlying gameplay, we use the `Pyro` probabilistic programming library (Bingham et al., 2019). Pyro provides the primitives required to implement SCMs consistent with the game's causal DAG, including stochastic functions for exogenous variables, deterministic assignments for endogenous variables, and intervention operators. This allows us to align the game engine's execution trace with an explicit causal model, and to sample parallel-world contrasts in a principled manner.

### K.3 REPRODUCIBILITY.

Both components are integrated in a unified Python codebase, ensuring that gameplay, causal modeling, and data generation can be run deterministically from a single seed.

### K.4 GRAPH LIBRARIES.

We generate mDAGs, parallel-world graphs and counterfactual graphs using the `Y0` library (Hoyt et al., 2025).

### K.5 GRAPH SERIALIZATION.

All graphs are serialized as directed graphs in JSON using a node-link format. Nodes are represented as JSON objects with keys for the node identifier and attributes, while edges are represented as objects with source and target identifiers and an edge type field. In the case of mDAGs and parallel world graphs, exogenous nodes are marked with the attribute `"exogenous": true`. For example, the above example of an mDAG with observed nodes $\{A, B, C\}$, one directed edge $A \rightarrow B$, and one hyper-edge $\{B, C\}$, is converted to a CL-DAG and serialized as follows:

```
{
  "nodes": [
```

```
    {"id": "A"},
    {"id": "B"},
    {"id": "C"},
    {"id": "N_{B,C}", "exogenous": true}
  ],
  "links": [
    {"source": "A", "target": "B", "type": "directed"},
    {"source": "N_{B,C}", "target": "B", "type": "directed"},
    {"source": "N_{B,C}", "target": "C", "type": "directed"}
  ]
}
```

## L    GENERALIZATION BEYOND GAMES

Our causal-consistency formulation applies wherever structured state, deterministic updates, and seed-controlled stochasticity are available—including physics simulators and VFX pipelines built on graphics engines. In partially controlled environments, one could generate a limited number of explicit contrasts and design objectives that approximate seed consistency and structure alignment when only noisy or implicit multiverse structure is available. However, transferring these insights to more complex real-world domains will require significant future work: realistic simulators involve higher-dimensional state and stochastic rendering that may not align cleanly across worlds, and real-world video lacks explicit multiverse structure entirely.

