# OpenReview forum: "Multiverse Mechanica: A Testbed for Learning Game Mechanics via Counterfactual Worlds"
_ICLR.cc/2026/Conference — ICLR 2026 Poster_

### Official Review · Reviewer_4gTF · 2025-10-20

**Soundness:** 1
**Presentation:** 2
**Contribution:** 1
**Rating:** 2
**Confidence:** 4

**Summary:**

The paper employs the formalism of causal learning to formulate the problem of learning game mechanics from visual observations. The paper proposes a framework to describe how interventions on (discrete) game states alter the corresponding image representations under different data generation conditions. This includes an example domain to generate data, including the relevant causal modeling information. To evalute the approach a diffusion model is trained to predict the visual outcomes of causal interventions generated using the framework.

**Strengths:**

# originality
Provides more formal rigor on defining the task of learning game mechanics from observations compared to prior efforts. The causal learning model here is novel relative to prior formalizations.


# quality
Explicitly seeks to create a data generator for model training that matches the formalism.


# clarity
Introduces the case study domain with a worked example (Table 1).


# significance
Opens the path to investigating causal learning for game mechanics based on discrete states. This could be of interest to studies of world models and their learning and limitations. Also for learning world models for game playing agents from demonstration videos.

**Weaknesses:**

# quality
The experiments seem mismatched to the objective of formalizing causal learning. The model starts by framing the problem as learning from video, then reduces that to a few input frames. Unfortunately, the evaluation metrics do not touch on the central question of whether the generated data was neccessary or sufficient for causal learning. Under questions I offer some suggestions on what experiments could help directly test this core idea. With development I believe these would be a strong launching point for the ideas to see further development and extension.

# clarity
The notation often was confusing and cluttered, rather than clarifying on the formalism. Perhaps it would help to have some listing of the symbols used and their definitions for easy reader reference.

The causal hierarchy being adopted should be introduced as background to readers. It's a core part of the framework, but never directly explained.

**Questions:**

# questions
- How are causal hierarchy level-1, level-2, and level-3 defined?
- How does the experiment demonstrate the need for level-3 data? (claims in lines 178-182)
- How does the experiment demonstrate generalization? (claims in lines 92-93)


# suggestions
- The following experiments may be a natural progression to test the claim that the level-2 and level-3 data are required for learning game mechanics. Given the established criteria of how well the model predicts outcomes given information on the intervention:
	- As a base experiment, remove all visual information (or map it as a single pixel indicating each state) and demonstrate the core claims. This setting decouples model representational learning capacity from causal learning. Then demonstrate that the model trained with equivalent amounts of level-1, level-2, and level-3 data shows the expected pattern of succeeding in tests only with level-3 and not the prior levels. Perhaps include a small scaling test for samples required.
		- The outcomes can easily be validated automatically without human evaluation. This bypasses the problems of assessment and directly tests the core claims.
	- Add the current scenario and evaluate on the above metrics.
	- Add the ability to generate scenarios that violate the assumptions required, like (fully or partially) hiding visual representation of the relevant causal variables.
	- Add generation of full videos and require learning on those to support the claim in the formalism around video input.
	- Add procedural variation of assets and placements to the framework to test generalization.

---

> ### Author Response · Authors · 2025-11-21
> **Clarifying causal identifiability**
>
> ### Relationship between experiments and the causal learning objective
>
> We appreciate your detailed comments and agree that connecting the experiments more tightly to causal identifiability is important. As discussed in our responses to the other reviewers, we are adding a supervised evaluation in which generated counterfactual frames are compared directly to ground-truth counterfactual frames rendered by the game engine. This directly assesses whether the learned model respects the counterfactual relations encoded by the mechanics.
>
> We can also view each mechanic statement \(S_i\) as a counterfactual query on the causal graph and ask whether it is identified from the training data. For example, for one component of the shield mechanic we consider
> \[
> S_1:\quad P(S_{W=0} = 0,\; S_{W=1} = 1) > \epsilon_1,
> \]
> where \(S\) indicates “shield equipped” and \(W\) denotes weapon weight.
>
> If \(S\) were observed under both interventions, training data would contain tuples
> \(\{s_i, s'_i, v_i, v'_i\}\) under \(do(W{=}0)\) and \(do(W{=}1)\). In that fully observed setting, standard graphical-identification arguments tell us that an empirical estimator over \((s_i,s'_i)\) converges to \(P(S_{W=0} = 0, S_{W=1} = 1)\).
>
> The case when \(S\) is *not* directly observed and must instead be inferred only from tuples of \(\{v_i, v'_i\}\) corresponds to learning a latent causal representation of the \(\{s_i, s'_i, v_i, v'_i \}\). Causal representation learning is a nontrivial task. However, we focus on a key desideratum of such a representation: that the generated values of \(V_{W=0}\) and \(V_{W=1}\) are **causally consistent**, i.e., differences between them are attributable only to the consequences of \(do(W{=}0)\) vs. \(do(W{=}1)\) across multiverse worlds. Our experiments evaluate the consistency of generated frames across worlds and against engine-generated ground truth as a practical proxy for the underlying causal query. We will clarify this connection and soften any language suggesting that we fully solve latent identifiability.
>
> ### Causal hierarchy and claims about level-3 data and generalization
>
> We will add a concise background table defining the three levels of the causal hierarchy (associational, interventional, counterfactual) and clarify that Multiverse Mechanica is explicitly designed to give us level-3 data via seed-locked multiverse worlds (parallel worlds sharing exogenous noise but differing in interventions). This is what allows us to pose mechanics as counterfactual queries at the model level.
>
> We agree that our initial wording may have overstated what our experiments show about the *necessity* of level-3 data. In the revision we will instead emphasize that:
>
> - the framework is constructed so that such questions can be posed and empirically studied; and
> - systematic scaling studies—varying observational/interventional/counterfactual data, hiding parts of the visual state, generating longer videos, and adding procedural variation—are natural next steps that the environment enables but that we do not fully explore in this first paper.
>
> ### Additional suggested experiments
>
> We are grateful for your concrete suggestions; they align well with our longer-term goals. Some aspects are already incorporated in spirit:
>
> - **Multiple timepoints.** We now evaluate at multiple labeled timepoints within an episode (turn start, action start, divergence, impact, convergence, etc.) rather than only at the impact frame, probing temporal cross-world consistency even though we use a frame-based diffusion model.
> - **Procedural variation.** The current environment already supports procedural variation in placements and asset layouts; we will make this more explicit and discuss how increasing this variation can be used to test generalization.
>
> We fully agree that additional experiments—such as training full video models and scaling to larger procedurally varied datasets—are the next logical steps in this research. Full large-scale video experiments are beyond what we can include here, but we will explicitly highlight these directions as natural extensions enabled by Multiverse Mechanica.
>
> ### Clarity and notation
>
> We take your point that the notation can feel cluttered. As mentioned in our responses to the other reviewers, we will:
>
> - add a notation table summarizing the main symbols used in the causal formalism;
> - simplify the core example in Section 3 to a single mechanic, with more complex cases moved to the appendix; and
> - explicitly introduce the causal hierarchy in the main text, with a pointer to our appendix primer.
>
> We hope these changes will make the formalism easier to follow while preserving the rigor you identified as a strength.

---

> > ### Comment · Reviewer_4gTF · 2025-11-28
> >
> > Thank you for the detailed responses. These are all valuable for improving the work.

---

### Official Review · Reviewer_WHRQ · 2025-10-25

**Soundness:** 3
**Presentation:** 4
**Contribution:** 3
**Rating:** 6
**Confidence:** 3

**Summary:**

The authors tackle the relevant but understudied problem of determining whether learnable models can correctly learn game mechanics. To do so, the authors contribute Multiverse Mechanica, a video game containing simple visuals, but reasonably complex game mechanics, which could serve as a benchmark for game mechanics learning. Defining characteristics of Multiverse Mechanica include its logic formalism for precisely describing game mechanics, and its ability to mark "impact frames", a sparse set of video frames containing full information on game mechanics which provide a convenient way to perform cheap experimentation. A latent diffusion model is trained on data produced by the synthetic environment, leveraging the established logical formalism to improve model understanding of game mechanics, but evaluation makes it difficult to determine whether this had positive impact on the model. The paper is well written and explains Multiverse Mechanics in great detail. Overall Multiverse Mechanica has potential to serve as a benchmark for game mechanics learning and the logical formalism introduced in the work could spark new ideas in the field, though it remains unclear if employment of such logical formalism will become applicable to more complex models and scenarios of practical interest.

**Strengths:**

- The paper tackles the relevant but not yet well studied problem of studying the degree to which generative models are capable of understanding game mechanics. In the rise of powerful video generation models investigation of this hard-to-achieve capability can become an interesting research direction.
- The authors contribute a videogame that was carefully designed to probe understanding of relatively complex game mechanics under constrained training budgets. The game is reasonably complex from the perspective of the contained mechanics, requiring reasoning on spell types, elemental affinities, weapon classes and equipment to make decision on taken damages. At the same time, its visual simplicity and ability to produce "impact frames" make it convenient for small scale ablations. This can constitute a good benchmark or tool for ablations for future works.
- The paper seems polished and detailed. Writing is clear and the appendix with its 45 total pages describes in great detail the proposed system and logic framework.

**Weaknesses:**

- The logic formalism employed in the paper may be unfamiliar to a large portion of readers. The authors make every effort to introduce the formalism, but the paper might not easily accessible to the broad generative models audience.
- The employed logic formalism appears difficult to define even for a relatively simple game as the one created by the authors. A concern is whether the proposed formalism and methodology could eventually scale to modeling the more complex game mechanics encountered in mainstream video game titles. If such possibility is not foreseen, the impact of the method could be limited. I could see a use for the formalism in evaluating in a rigorous way video sequences for the purpose of determining if a model has learned the described mechanic correctly, but I don't see an easy way of scaling training as performed in Sec 5 on a more complex scenario.
- The proof of concept model shown in Sec. 5 is built based on the assumption that contrasts are known, a condition that is satisfied for the proposed synthetic data generator. For example, it proposes a seed consistency loss aligning noise for samples across contrasts and a structure alignment loss. These components would be difficult to leverage in the context of large scale video generation methods, which would be the ultimate models on which to perform evaluation for game mechanics understanding, but that would require large amounts of data for which contrasts would be very difficult to derive.
- The experimental setting with respect to employed metrics and experimental setting is unclear, making it difficult to interpret results. I suggest authors to describe each of the employed metrics and its computation procedure.
- Interpretation of the experimental results is difficult because of the lack of baselines or ablations. The paper does not validate whether any of the introduced components in Sec 5 improves model performance with respect to straightforward diffusion model training.

**Questions:**

I'd like authors to clarify the aspects highlighted in the weaknesses section with particular regards to the concerns regarding clearer definition of experimental setting, definition of metrics, and creation of a baseline against which to compare the proposed method's performance.

---

> ### Author Response · Authors · 2025-11-21
> **Clarifying the formalism, scalability, and empirical validation of Multiverse Mechanica**
>
> ### Accessibility of the logic formalism
>
> We appreciate your recognition of the formalism’s potential and your concern about accessibility. In the revision, we will:
>
> - add a brief, intuitive overview in the main text;
> - provide a compact notation table and a simplified running example in Section 3, moving more complex variants to the appendix;
> - include a short concepts box for the three levels of the causal hierarchy in the game setting, with a clearer pointer to the appendix primer.
>
> These changes are intended to make the paper easier to follow for readers less familiar with causal theory.
>
> ### Scalability to more complex games and models
>
> We agree that encoding AAA mechanics in a formal causal language is nontrivial. In the worst case, one could specify a causal DAG for a game much as one does for real-world domains (economics, epidemiology, etc.). AAA games are still simpler than reality and easier to probe with experiments; when code is available, a DAG can also be derived by analyzing causal dependencies in the game logic (e.g., damage pipeline). We see this large-scale analysis as important future work but beyond the scope of this paper.
>
> We will emphasize that Multiverse Mechanica is meant as a controlled benchmark and proof-of-concept: it shows that SCM-style vocabulary can describe game mechanics and guide data/objective design. We will also clarify that more complex mechanics can be added progressively and that existing design documents can serve as a practical starting point when scaling up.
>
> ### Use of known contrasts and applicability to large-scale video models
>
> You are correct that our Multiverse Alignment Loss relies on known contrasts between seed-locked multiverse pairs, which are naturally available in our synthetic environment. This is deliberate: the environment is designed to study what becomes possible when contrasts are explicit.
>
> In the revision we will briefly discuss how similar ideas might transfer to more realistic settings, e.g., by generating a limited number of explicit contrasts in engines or VFX pipelines we can partially control, and by designing objectives that approximate seed-consistency and structure alignment when only noisy or implicit multiverse structure is available. This positions Multiverse Mechanica as a “best-case” benchmark whose insights can inform large-scale video models trained under weaker supervision.
>
> ### Experimental setting and metric definitions
>
> As also noted in response to Reviewer LygS, we will add explicit definitions for each metric and group them as:
>
> 1. **Semantic consistency metrics** (CF Transfer CLIP, Exogenous Distance);
> 2. **Image and ground-truth fidelity metrics** (PSNR/SSIM);
> 3. **Null-intervention reconstruction metrics** (reconstructing the factual under no mechanic change);
> 4. **Minimality-of-edits metrics** (how selectively the mechanic changes the scene).
>
> To clarify interpretation, we have introduced a supervised evaluation regime in which generated counterfactual frames are compared directly to ground-truth counterfactual frames rendered by the game engine under the same \((\omega, X, C)\). Concretely, we compute the same PSNR/SSIM-style metrics both (i) between factual and counterfactual images and (ii) between generated counterfactuals and engine counterfactuals (how well the model recovers the true counterfactual).
>
> ### Baselines and ablations
>
> We agree that ablations and baselines are needed. We have implemented ablations removing the seed-consistency term (L1) and the structure-alignment term (L2) separately, as well as naive diffusion fine-tuning and a CLIP-based contrastive baseline, all trained on the same Multiverse Mechanica data. Due to the rebuttal timeline, we currently report results on a held-out subset and will extend all metrics to the full test set in the revised manuscript.
>
> On this subset, comparing the full Multiverse Alignment Loss (L1+L2) to the abduction-only variant (w/o L2) shows a consistent pattern: CF Transfer CLIP is markedly higher with L1+L2 (≈24 vs. ≈17); L1+L2 trades a small drop in reconstruction PSNR for clearly higher SSIM and CLIP scores; factual–counterfactual image quality is essentially unchanged; and when comparing generated counterfactuals to engine counterfactuals, the aligned model achieves much better structural agreement (higher SSIM), even when PSNR is slightly lower. Exogenous Distance (MSE between abducted exogenous seeds) remains very small in all settings, as expected from seed-locked pairs.
>
> Metrics for the L1-only ablation, naive fine-tuning, and CLIP-based baselines are being computed with the same protocol. In the revised manuscript we will present full test-set results and summarize the overall pattern: the full Multiverse Alignment Loss most strongly improves causal-consistency metrics while maintaining competitive image and reconstruction quality, supporting the practical value of the logically motivated components in Section 5.

---

> > ### Comment · Reviewer_WHRQ · 2025-11-24
> >
> > I thank the authors for their answer. The answer addresses some of my main concerns that seem shared across reviewers:
> > - Accessibility of the paper and clarity of presentation
> > - Explanation of metrics and introduction of metric directly comparing against ground truth counterfactual frames
> > - Introduction of ablation experiments
> >
> > Regarding the point of applying the proposed methodology to more complex or real-world scenarios, I think it makes sense to consider Multiverse Mechanica as a "best-case" benchmark and starting point for studying weaker forms of supervision, though applying learnings from this setting to more complex or real-world cases may require significant amount of future work. Showing an example where this is achieved would have strengthened the work. As in my original review, I think Multiverse Mechanica will serve as a useful tool for the community and as a starting point to perform such future work, and I will keep my acceptance recommendation.

---

> > > ### Author Response · Authors · 2025-11-26
> > > **From Multiverse ITEs to CATEs**
> > >
> > > We thank the reviewer for their feedback. We agree with the view that this work isa starting point for weaker supervision in future work where simulating multiverse outcomes is not feasible but approximating them is. In causal inference terms, this is analogous to substituting an individual treatment effect with a conditional average treatment effect conditioned on many attributes of the individual. Indeed, this is a practical and promising direction for future work.

---

### Official Review · Reviewer_LygS · 2025-10-30

**Soundness:** 3
**Presentation:** 2
**Contribution:** 3
**Rating:** 6
**Confidence:** 2

**Summary:**

The authors present a new game environment designed to test the ability for pixel-based game world models to accurately learn game mechanics. The environment is designed to automatically produce causal graphs that represent specific game mechanics to allow for ground-truth checking against model predictions. The authors train a proof-of-concept model in their proposed environment and argue that it points the way towards richer training of other world models.

**Strengths:**

I think the motivation of this paper is quite compelling. The introduction does a good job both laying out existing perspectives on the nature of pixel-based world models and arguing for them to be made more rigorous and formal. In general, the introduction does a good job of laying out why the paper's contributions would be both novel and timely. The Multiverse Mechanica game, while not currently available, does also seem like a useful contribution to other researchers interested in studying world models. While the technical foundations presented in the paper are at times a bit confusing (see below), they also point towards the authors' deep understanding of the  subject matter.

**Weaknesses:**

I think the paper suffers somewhat in terms of clarity. I admit that this may be due to my own unfamiliarity with the relevant mathematical underpinnings, but I found the technical descriptions in Section 3 fairly difficult to follow and the accompanying figures too busy to easily parse. Later sections use a variety of evaluation methods without clearly defining them (e.g. exogenous distance and transfer CLIP), which makes it difficult to clearly understand the implications of the proof-of-concept experiment. In particular, I wonder about the claim that “the model successfully generates semantically meaningful counterfactuals” -- this seems like something that would benefit greatly from one or more concrete example outputs. In the absence of any baseline models against which to compare or a dedicated user study to evaluate the outputs, it’s difficult to say what the model has accomplished.

**Questions:**

- How should we interpret the quantitative results in Table 3? More specifically, what might an unsuccessful model and an oracle model look like?
- Line 161: is this meant to say the rows of Table 1 instead of columns?
- For S3 in Table 1: why is it necessary to condition S = 1? Are there ways for a player to block aside from having a shield, and does blocking have different behavior in that context?

---

> ### Author Response · Authors · 2025-11-21
> **Clarifying the formalism and metrics, and adding qualitative examples and baselines for Multiverse Mechanica**
>
> ### Clarity of Section 3 and figures
>
> Thank you for highlighting the clarity issues. We will improve Section 3 along three axes:
>
> - Add a concise notation table/glossary summarizing the main variables and operators used in the formalism.
> - Simplify the main counterfactual graph figure by focusing on a single running example mechanic and moving more detailed variants to the appendix.
> - Reorganize the exposition to more clearly connect each mathematical component to the running example in the game.
>
> We believe these changes will make the formalism more accessible to readers who are not already familiar with causal notation.
>
> ### Metric definitions and interpretation of Table 3
>
> We agree that several metrics were insufficiently defined in the main text. We will:
>
> - Explicitly define *CF Transfer CLIP* as CLIP similarity between generated counterfactual images and the target mechanic description.
> - Define *Exogenous Distance* as the mean squared error between abducted seeds \(\hat\omega_0,\hat\omega_1\) for a multiverse contrast.
> - Add a short subsection that groups metrics into: (i) causal-consistency metrics, (ii) image \& ground-truth fidelity metrics, (iii) composition under null-intervention, and (iv) minimality-of-edits metrics, with intuitive descriptions.
>
> Intuitively, our quantitative results can be interpreted as follows. A model would be judged to have *not* learned a mechanic if, given an intervention \(X{=}x\) and a game-generated frame in world A \(V_{X=x}\), and an intervention \(X{=}x'\) in world B with corresponding game-generated frame \(V'_{X=x'}\), it fails to generate a frame in world B, \(\hat{V}'_{X=x'}\), that is different in ways not attributable to the differences between the consequences of \(X{=}x\) and \(X{=}x'\) (minimal edits). Also, \(\hat{V}'_{X=x'}\ should be similar to \(V'_{X=x'}\) (correct counterfactual label). Metrics like PSNR and SSIM are standard for evaluating similarity in image editing; they quantify whether an edit is minimal. In our setting, a "minimal" implies consistency -- we can view \(\hat{V}'_{X=x'}\) as an edit of \(V_{X=x}\); if \(\hat{V}'_{X=x'}\) differs from \(V_{X=x}\) only in mechanic-relevant regions while remaining similar to the engine-generated \(V'_{X=x'}\), then the edit is both minimal and correct. We will clarify this interpretation and the role of each metric in the revision.
>
> ### Semantically meaningful counterfactuals and examples
>
> We agree that the claim that “the model successfully generates semantically meaningful counterfactuals” benefits from concrete illustrations. In the revised manuscript, we will add qualitative panels showing factual/counterfactual pairs for multiple mechanics, highlighting how:
>
> - global layout and exogenous details (background, camera, non-affected entities) are preserved across worlds; and
> - only mechanic-dependent aspects (e.g., presence of shield, type of damage effect) are toggled between worlds.
>
> We will select examples from the newly trained models underlying our updated metrics (described in the response to Reviewer vVfQ), including cases where baselines visibly fail to maintain exogenous consistency, to make the quantitative differences more intuitive.
>
> ### Baselines and user study
>
> You are right that our initial submission lacked baselines, which made it harder to assess what the model has accomplished. As described in our response to Reviewer vVfQ and WHRQ, we have now added ablations (removing L1 and L2 separately) and simple baselines (naive fine-tuning and a CLIP-only contrastive alignment variant) on the same Multiverse Mechanica data. Our rebuttal summarizes intermediate results on a held-out subset of test pairs; the revised manuscript will include full-test-set numbers and standard errors.
>
> Given the strong supervision available from the engine (and the precise mechanistic labels), we prioritized automatic, engine-grounded metrics over a user study in this first work. We will state this explicitly and position human evaluation of perceived mechanic correctness as complementary future work that can build on the benchmark we introduce here.
>
> ### Other questions
>
> **(a) Line 161: rows vs. columns.**
> Thank you for catching this. This is indeed a typo: the text should refer to the *rows* of Table 1, not “the columns of Figure 1”. We will correct this in the revision.
>
> **(b) S3 conditioning on \(S=1\).**
> In the current game design, it is possible to successfully block without a shield (but the probability of taking damage is much higher in that case), and it is also possible to fail to block despite having a shield. To isolate the shield mechanic, S3 conditions on \(S{=}1\) to focus specifically on block events mediated by an equipped shield. We will clarify this design choice and its implications for the mechanic query in the text.

---

### Official Review · Reviewer_vVfQ · 2025-10-31

**Soundness:** 2
**Presentation:** 3
**Contribution:** 3
**Rating:** 4
**Confidence:** 3

**Summary:**

The paper proposes a causal framework for making generative models mechanically consistent (that is, able to change only the causal effects of an intervention while keeping all else fixed). To this end, the authors introduce Multiverse Mechanica, a controllable game environment that generates counterfactual video pairs: parallel worlds sharing the same exogenous randomness (non-descendant variables) but differing in a single game mechanic (e.g., shield on/off). Using single impact frames from these videos, they finetune a pretrained T2I diffusion model (OpenJourney-v4) with a new Multiverse Alignment Loss that enforces (a) shared exogenous seed consistency and (b) early-stage structural alignment. Experiments show that the finetuned model preserves global scene structure while correctly toggling mechanic-specific effects, demonstrating a step toward causally consistent generative modeling.

**Strengths:**

- The paper introduces a causally principled definition of mechanical consistency via counterfactual constraint. Unlike previous works which focused on visual realism, it provides a framework of grounding generative modeling in causal reasoning.

- The Multiverse Mechanica environment provides a controllable, reproducible setting with groundtruth causal graphs and consistent counterfactual pairs. This is valuable for studying causal generalization in generative models. (Although the code and dataset release is remained to be seen.)

- The proposed Multiverse Alignment Loss (seed consistency & structure alignment) is intuitive and conceptually sound. It aligns neatly with the diffusion process’s structure (e.g., early steps for global layout; late steps for fine details).

- In their proof-of-concept experiment, qualitative and quantitative results show that the finetuned diffusion model preserves global scene structure while correctly toggling mechanic-dependent effects, supporting the causal-consistency claim.

**Weaknesses:**

- The approach is tested only on _single-frame_ impact images, not full temporal sequences. This prevents evaluation of whether the proposed causal consistency generalizes across time, although the authors claim their framework should, in principle, extend to video models.

- All results focus on a single mechanic example ("shield on/off") and one pretrained model (OpenJourney-v4). It remains unclear whether the framework scales to multiple simultaneous mechanics or to other base models. Again, this should work in principle, but empirical evidence is still lacking.

- In terms of empirical validation, the paper also lacks ablation and baseline comparisons. Since the new loss formulation is one of the main contributions, it would be highly beneficial to show what happens when either the seed-consistency (L1) or structure-alignment (L2) term is removed, and to compare against simpler alternatives (for example, standard fine-tuning or contrastive alignment).

- Finally, the evaluation metrics are indirect. While CLIP, PSNR, and SSIM provide useful proxies, they do not directly capture causal correctness. The paper could propose or adopt explicit causal evaluation metrics to better quantify causal consistency.

**Questions:**

See the above weaknesses.

Besides, I have one question (or suggestion): do you think your framework could be applied beyond the game domain? It seems the same causal-consistency formulation could extend to other areas such as physics simulation or movie generation. If so, adding a short paragraph discussing these broader applications or potential impacts could help widen the paper’s relevance and readership.

---

> ### Author Response · Authors · 2025-11-21
> **Supervised causal metrics, temporal analysis beyond impact frames, and ablations of the multiverse alignment loss**
>
> ### Metrics for causal consistency in image and video
>
> We agree that CLIP, PSNR, and SSIM are *indirect* proxies for causal consistency the emphasize minimizing differences between a factual frame and a generated counterfactual frame. In response to your feedback, we expanded the experiments to the generated counterfactual frames to game-generated ground-truth counterfactual frames. On a held-out subset of test pairs, current runs yield CF-vs-GT scores of about 9.1 dB PSNR and 0.21 SSIM for our full multiverse objective, and about 11.4 dB PSNR and 0.08 SSIM for the ablation without L2. The ablation slightly improves PSNR but substantially worsens structural similarity, supporting our view that the alignment term trades some pixel-level PSNR for structural and semantic faithfulness. We will report full test-set numbers in the revised manuscript.
>
> ### Temporal structure and impact frames
>
> We agree that focusing only on the time of peak action ("impact frames") does not fully address temporal causal consistency. Yet, a principled metric for temporal causal consistency in video pairs remains an open problem. The ideal (beyond our scope) would be a video language model that could rigorously verify if the differences between two multiverse videos were attributable only to the differing interventions. Our game approximates this by labeling key time points during an episode and allowing comparison of individual frames with shared labels across a multiverse pair. These labels include the aforementioned time of peak action, as well as turn start/end times, action/reaction start times, when videos in a multiverse pair first diverge when they again resynchronize. In the proof-of-concept experiment we used only impact frames. In response to your comment, we are rerunning the analysis over all labeled timepoints, computing CLIP/PSNR/SSIM per label and averaging across labels. On the subset evaluated so far, qualitative behavior and trends match our original finding that the multiverse objective improves cross-world consistency. We will add this extended analysis and explicitly note that our current evaluation is limited to settings with similar temporal structure.
>
> ### Single mechanic
>
> Our dataset and training setup in fact span all implemented mechanics; the “shield on/off” example was chosen because it is easy to visualize. We will state that training covers the full mechanic set (shield, elemental immunity, weapon range, and spell-related submechanics) and add qualitative examples for multiple mechanics in the appendix.
>
> ### Ablations and baselines
>
> We appreciate the request for ablations. We have implemented ablations without the seed-consistency term (L1) and without the structure-alignment term (L2), as well as naive fine-tuning and a CLIP-space contrastive baseline. Due to the rebuttal timeline, we currently report results on a held-out subset and will extend all metrics to the full test set in the revised manuscript.
>
> On this subset, comparing the full Multiverse Alignment Loss (L1+L2) to the abduction-only variant (w/o L2) shows:
>
> - **Causal consistency.** CF Transfer CLIP is much higher with L1+L2 (≈24 vs. 17), indicating that counterfactual images are more semantically aligned with the target mechanic.
> - **Reconstruction vs. structure.** Removing L2 yields higher reconstruction PSNR, but L1+L2 achieves markedly higher SSIM and reconstruction CLIP. The alignment term trades some pixel-level PSNR for structural and semantic fidelity, which is closer to our causal objective.
> - **Base image quality and CF–GT agreement.** Factual–CF similarity metrics are essentially identical between L1+L2 and w/o L2, suggesting that alignment does not degrade global image quality or the magnitude of the mechanic-induced change. When comparing generated CFs to engine CFs, w/o L2 has slightly higher PSNR, while L1+L2 has much higher SSIM, indicating better structural agreement with the true counterfactual.
>
> Exogenous Distance (MSE between abducted exogenous seeds) is very small in both settings, as expected under seed-locked pairs. Metrics for the L1-only ablation, naive fine-tuning, and CLIP-contrastive baselines are being computed with the same protocol and will be reported for the full test set in the revised manuscript.
>
> ### Additional model backbone
>
> To assess robustness to the base model, we are also finetuning Stable Diffusion 1.5 under the same Multiverse Alignment Loss, to check that the observed trends are not specific to OpenJourney-v4. Due to the rebuttal timeline, we focus here on the OpenJourney-v4 results and will include Stable Diffusion 1.5 results in the revised manuscript.
>
> ### Applicability beyond games
>
> We appreciate the suggestion to broaden the discussion. The key ingredients of our approach also apply to other controlled simulators and to VFX pipelines built on graphics engines. We will add a brief paragraph outlining these broader applications.

---

### Meta-Review · Area_Chair_EZDa · 2025-12-16

**Summary:**

This work introduces Multiverse Mechanica, a causal testbed game that generates consistency-guaranteed counterfactual data to evaluate whether generative models can learn underlying game mechanics. Four reviewers carefully reviewed this paper, with two marginally above the acceptance threshold (rating 6), one marginally below the acceptance threshold (rating 4), and one rejection (rating 2).

The authors wrote a thorough response to address the reviewers’ concerns. One of the limitations is that although the reviewers said that they would add revisions to the manuscript, it is a bit difficult to locate the details in the submitted revision. Actually, most of the parts have not yet been revised. But in my view, most of the concerns have already been addressed. Concretely:

Reviewer vVfQ asked about the metrics, baselines, and ablation studies. The authors have carefully responded to these questions. The authors also promised to add more examples and discussions, e.g., additional examples beyond "shield on/off" and a discussion of applicability beyond games.

Reviewers LygS and WHRQ both gave positive ratings. And most of their concerns have been solved.

Although Reviewer 4gTF gave negative ratings, I think his reasons for rejection are not strong enough. During the rebuttal, the authors have clarified the details of symbols and experiments. Reviewer 4gTF responded in the rebuttal and did not raise additional concerns.

In my view, this paper can be accepted. I hope the authors can carefully revise the manuscript as they promised in the rebuttal.

**Reviewer Concerns:**

Most of the concerns have already been addressed.

**Reviewer Scores:**

Reviewer 4gTF may improve his rating.

---

### Decision · Program_Chairs · 2026-01-26

Accept (Poster)